# GRADIENT NORM AS A POWERFUL PROXY TO OUT-OF-DISTRIBUTION ERROR ESTIMATION

## ABSTRACT

Estimating out-of-distribution (OOD) error without access to the ground-truth test labels is a challenging, yet extremely important problem in the safe deployment of machine learning algorithms. Current works rely on the information from either the outputs or the extracted features to formulate an estimation score correlating with the expected OOD error. In this paper, we investigate–both empirically and theoretically–how the information provided by the gradients can be predictive of the OOD error. Specifically, we use the norm of classification-layer gradients, backpropagated from the cross-entropy loss after only one gradient step over OOD data. Our key idea is that the model should be adjusted with a higher magnitude of gradients when it does not generalize to the OOD dataset. We provide theoretical insights highlighting the main ingredients of such an approach ensuring its empirical success. Extensive experiments conducted on diverse distribution shifts and model structures demonstrate that our method outperforms state-of-the-art algorithms significantly.

## 1 INTRODUCTION

Deploying machine learning models in the real world is often subject to a distribution shift between training and test data. Such a shift may significantly degrade the model's performance during test time (Quinonero-Candela et al., 2008; Geirhos et al., 2018; Koh et al., 2021) and lead to high risks related to AI satefy (Deng & Zheng, 2021). To alleviate this problem, a common practice is to monitor the model performance regularly via collecting the ground truth of a subset of the current test dataset (Lu et al., 2023). This practice, however, is usually time-consuming and expensive highlighting the importance of proposing unsupervised methods for estimating test performance under distribution shift, commonly known as *OOD error estimation*.

**Limitations of current approaches.** Current studies mainly focus on outputs or feature representation to derive an ODD error estimation score. Such a score represents calibrated test error or distribution discrepancy from the training to the test datasets (Hendrycks & Gimpel, 2016; Guillory et al., 2021; Garg et al., 2022; Deng & Zheng, 2021; Yu et al., 2022b; Lu et al., 2023). For example, Hendrycks & Gimpel (2016) calculates the average maximum softmax score of the test samples as the estimated error. Similarly, Garg et al. (2022) learns a confidence threshold from the training distribution. Deng & Zheng (2021) quantifies the distribution difference between the training and the test datasets in the feature space, while Yu et al. (2022b) gauges the distribution gap in the parameter level. Although insightful, existing methods are either prone to overfitting (Wei et al., 2022), have a high computational cost (Yu et al., 2022b), or require strong assumptions about the underlying distribution shift (Lu et al., 2023). To address those issues, we propose a lightweight, robust, and strongly predictive OOD error estimation method relying on gradients that were often overlooked in prior work. Inspired by the strong correlation of the gradients with the generalization error of deep neural networks (Li et al., 2019; An et al., 2020), we seek to answer the following question:

*Are gradients predictive of OOD errors?*

**Our contributions.** We tackle this question by transposing the idea of using the gradient norm for generalization error prediction to the OOD setting. We hypothesize that the model requires a large magnitude gradient step if it cannot generalize well on the OOD dataset at hand. To quantify the

magnitude of gradients, we propose a simple yet efficient gradient-based statistic, GRDNORM *Score*, which employs the vector norm of gradients backpropagated from a standard cross-entropy loss on the test samples. To avoid the need for ground-truth labels, we propose a pseudo-labeling strategy that seeks to benefit from both correct and incorrect predictions. Ultimately, we demonstrate that the simple one-step gradient norm of the classification layer strongly correlates with the generalization performance under diverse distribution shifts acting as a strong and lightweight proxy for the latter.

The main contributions of our paper are summarized as follows:

1. We first provide several theoretical insights showing that correct pseudo-labeling and gradient norm have a direct impact on the OOD error estimation. This is achieved by analyzing the analytical expression of the gradient after 1 gradient step over the trained model on OOD data and by upper-bounding the population OOD risk.

2. Based on these theoretical insights, we propose the GRDNORM Score, which gauges the magnitude of the classification-layer gradients and shows a strong correlation with OOD error. Our method does not require access to training datasets and only needs one step of backpropagation making it particularly lightweight in terms of computational efficiency.

3. We perform a large-scale empirical evaluation of GRDNORM showing its superior performance. We achieve new state-of-the-art results on 11 benchmarks across diverse distribution shifts compared to 8 competitors, while being faster than the previous best baseline.

The rest of the paper is organized as follows. Section 2 presents the necessary background on the problem setup of our work. In Section 3, we derive several theoretical insights that motivate the GRDNORM Score introduced afterward. Section 4 is devoted to extensive empirical evaluation of our method, while ablation study is deferred to Section 5. Finally, Section 6 concludes our work.

## 2 BACKGROUND KNOWLEDGE

We tackle the problem of *OOD error estimation* which is of great practical interest when learning under distribution shifts and no ground-truth labels of test samples. The goal is to estimate the test classification accuracy of a specific OOD dataset using unlabeled test samples and/or labeled training datasets (see Appendix B for more details). To the best of our knowledge, our work is the first to study the linear relationship between the gradients and the model performance for this learning task.

**Problem setup.** We consider a $K$-class classification task with the input space $\mathcal{X} \subset \mathbb{R}^D$ and the label set $\mathcal{Y} = \{1, \ldots, K\}$. Our learning model is a neural network with trainable parameters $\boldsymbol{\theta} \in \mathbb{R}^p$ that maps from the input space to the label space $f_{\boldsymbol{\theta}} : \mathcal{X} \to \mathbb{R}^K$. We view the network as a combination of a complex feature extractor $f_{\mathbf{g}}$ and a linear classification layer $f_{\boldsymbol{\omega}}$, where $\mathbf{g}$ and $\boldsymbol{\omega} = (\mathbf{w}_k)_{k=1}^K$ denote their corresponding parameters. Given a training example $\mathbf{x}_i$, the feedforward process can be expressed as:

$$f_{\boldsymbol{\theta}}(\mathbf{x}_i) = f_{\boldsymbol{\omega}}(f_{\mathbf{g}}(\mathbf{x}_i)). \tag{1}$$

Let $\mathbf{y} = (y^{(k)})_{k=1}^K$ denote the one-hot encoded vector of label $y$, i.e., $y^{(k)} = 1$ if and only if $y = k$, otherwise $y^{(k)} = 0$. Then, given a training dataset $\mathcal{D} = \{\mathbf{x}_i, y_i\}_{i=1}^n$ that consists of $n$ data points sampled *i.i.d.* from the source distribution $P_S(\mathbf{x}, y)$ defined over $\mathcal{X} \times \mathcal{Y}$, $f_{\boldsymbol{\theta}}$ is trained following the empirical cross-entropy loss minimization:

$$\mathcal{L}_{\mathcal{D}}(f_{\boldsymbol{\theta}}) = -\frac{1}{n} \sum_{i=1}^n \sum_{k=1}^K y_i^{(k)} \log \mathrm{s}_{\boldsymbol{\omega}}^{(k)}(f_{\mathbf{g}}(\mathbf{x}_i)), \tag{2}$$

where $\mathrm{s}_{\boldsymbol{\omega}}^{(k)}$ denotes the softmax function for class $k$ that outputs an approximation of the posterior probability $P(Y = k|\mathbf{x})$: $\mathrm{s}_{\boldsymbol{\omega}}^{(k)}(f_{\mathbf{g}}(\mathbf{x})) = \exp\{\mathbf{w}_k^{\mathsf{T}} f_{\mathbf{g}}(\mathbf{x})\} / \left( \sum_{\tilde{k}} \exp\{\mathbf{w}_{\tilde{k}}^{\mathsf{T}} f_{\mathbf{g}}(\mathbf{x})\} \right)$.

**OOD error estimation.** We now assume to have access to $m$ OOD test samples $\mathcal{D}_{\text{test}} = \{\tilde{\mathbf{x}}_i\}_{i=1}^m \sim P_T(\mathbf{x})$. For each test sample $\tilde{\mathbf{x}}_i$, we predict the label by $\tilde{y}_i' = \arg\max_{k \in \mathcal{Y}} f_{\boldsymbol{\theta}}(\tilde{\mathbf{x}}_i)$. We now want to assess the performance of $f_{\boldsymbol{\theta}}$ on a target distribution without having access to ground-truth labels $\{\tilde{y}\}_{i=1}^m$ by estimating as accurately as possible the following quantity:

$$\mathrm{Err}(\mathcal{D}_{\text{test}}) = \frac{1}{m} \sum_{i=1}^m \mathbb{1}(\tilde{y}_i' \neq \tilde{y}_i), \tag{3}$$

where $\mathbb{1}(\cdot)$ denotes the indicator function. In practice, OOD error estimation methods provide a proxy score $S(\mathcal{D}_{\text{test}})$ that should exhibit a linear correlation with $\text{Err}(\mathcal{D}_{\text{test}})$. The performance of such methods is measured using the coefficient of determination $R_2$ and the Spearman correlation coefficient $\rho$. In what follows, we use the terms "source", "ID" and "train" interchangeably and similarly for "target", "test" and "OOD".

**Remark 2.1** (Differences with OOD detection). *The OOD error estimation problem should not be confused with the OOD detection problem. In a nutshell, the first one has to do with giving a proxy of the OOD test error, while the latter aims at classifying between in-distribution (ID) and OOD data points (see Appendix B for more details). In Appendix E, we show that a recent gradient-based OOD detection method (Huang et al., 2021) is inferior to our approach for OOD error estimation suggesting that the two problems cannot be tackled with the same tools.*

## 3 GRDNORM SCORE: THE ENDING IS DETERMINED IN THE BEGINNING

We start by deriving an analytical expression of the gradient obtained when fine-tuning a source pre-trained model on new test data. We further use the intuition derived from it to propose our OOD error estimation score, and justify its effectiveness through a more thorough theoretical analysis.

**A motivational example.** Below, we follow the setup considered by Denevi et al. (2019); Balcan et al. (2019); Arnold et al. (2021) to develop our intuition behind the importance of gradient norm in OOD error estimation. Our main departure point for this analysis is to consider fine-tuning: a popular approach to adapting a pre-trained model to new labeled datasets is to update either all or just a fraction of its parameters using gradient descent on the new data. To this end, let us consider the following linear regression example, where the test OOD data is distributed as $X \sim \mathcal{N}(0, \sigma_t^x)$, $(Y|X{=}x) \sim \mathcal{N}(\theta_t x, 1)$ parameterized by the optimal regressor $\theta_t \in \mathbb{R}$, while the data on which the model was trained is distributed as $X \sim \mathcal{N}(0, \sigma_s^x)$, $(Y|X{=}x) \sim \mathcal{N}(\theta_s x, 1)$ with $\theta_s \in \mathbb{R}$. Consider the least-square loss over the test distribution:

$$\mathcal{L}_T(c) = \frac{1}{2}\mathbb{E}_{P_T(x,y)}(y - cx)^2.$$

When we do not observe the target labels, one possible solution would be to analyze fine-tuning when using the source generator $(Y|X{=}x) \sim \mathcal{N}(\theta_s x, 1)$ for pseudo-labeling. Then, we obtain that

$$\begin{aligned}
\frac{1}{2}\nabla_c \mathbb{E}_{P_T(x)}\mathbb{E}_{P_S(y|x)}[(y - cx)^2] &= \mathbb{E}_{P_T(x)}\mathbb{E}_{P_S(y|x)}[(y - cx)(-x)] \\
&= \mathbb{E}_{P_T(x)}\mathbb{E}_{P_S(y|x)}[cx^2 - xy] \\
&= (c - \theta_s)\sigma_t^x \\
&= ((c - \theta_t) + (\theta_t - \theta_s))\sigma_t^x.
\end{aligned}$$

This derivation, albeit simplistic, suggests that the gradient over the target data correlates – modulo the variance of $x$ – with $(c - \theta_t)$, capturing how far we are from the optimal parameters of the target model, and $(\theta_s - \theta_t)$, that can be seen as a measure of dissimilarity between the distributions of the optimal source and target parameters. Intuitively, both these terms are important for predicting the OOD performance suggesting that the gradient itself can be a good proxy for the latter.

### 3.1 PROPOSED METHODS

We now formally introduce our proposed score, termed GRDNORM, to estimate OOD error during evaluation. We start by recalling the backpropagation process of the pre-trained neural network $f_\theta$ from a cross-entropy loss and then describe how to leverage the gradient norm for the OOD error estimation. The detailed algorithm can be found in Appendix A.

**Feedforward.** Similar to the feedforward in the pre-training process shown in Eq. 1, for any given test individual $\tilde{\mathbf{x}}_i$, we have:

$$f_{\boldsymbol{\theta}}(\tilde{\mathbf{x}}_i) = f_{\boldsymbol{\omega}}(f_{\mathbf{g}}(\tilde{\mathbf{x}}_i)). \tag{4}$$

As explained above, we do not observe the true labels of OOD data. We now detail our strategy for pseudo-labeling that allows us to obtain accurate and balanced proxies for test data labels based on accurate and potentially inaccurate model predictions.

**Label generation strategy.** Unconditionally generating pseudo-labels for OOD data exhibits an obvious drawback: we treat all the assigned pseudo-labels as correct predictions when calculating the loss, ignoring the fact that some examples are possibly mislabeled. Therefore, we propose the following confidence-based label-generation policy that outputs for every $\tilde{\mathbf{x}}_i \in \mathcal{D}_{\text{test}}$:

$$\tilde{y}'_i = \begin{cases} \arg\max_k f_{\boldsymbol{\theta}}(\tilde{\mathbf{x}}_i), & \max_k s_{\boldsymbol{\omega}}^{(k)}(f_{\mathbf{g}}(\tilde{\mathbf{x}}_i)) > \tau \\ \tilde{y}' \sim U[1, K], & \text{otherwise} \end{cases} \tag{5}$$

where $\tau$ denotes the threshold value, and $U[1, k]$ denotes the discrete uniform distribution with outcomes $\{1, \ldots, K\}$. In a nutshell, we assign the predicted label to $\tilde{\mathbf{x}}_i$, when the prediction confidence is larger than a threshold while using a randomly sampled label from the label space otherwise. The detailed empirical evidence justifying this choice is shown in Section 5 and we discuss the choice of proper threshold $\tau$ in Appendix F. From the theoretical point of view, our approach assumes that the classifier makes mistakes mostly on data with low prediction confidence, for which we deliberately assign noisy pseudo-labels. Feofanov et al. (2019) used a similar approach to derive an upper bound on the test error and proved its tightness in the case where the assumption is satisfied. We discuss this matter in more detail in Appendix G.

**Backpropagation.** To estimate our score, we calculate the gradients w.r.t. the weights of the classification layer during the first epoch backpropagated over the standard cross-entropy loss. The loss function can be written as follows:

$$\mathcal{L}_{\mathcal{D}_{\text{test}}}(f_{\boldsymbol{\theta}}) = -\frac{1}{m} \sum_{i=1}^{m} \sum_{k=1}^{K} \tilde{y}_i'^{(k)} \log s_{\boldsymbol{\omega}}^{(k)}(f_{\mathbf{g}}(\tilde{\mathbf{x}}_i)), \tag{6}$$

Given parameters of the classification layer $\boldsymbol{\omega}$, the gradients of the standard cross-entropy loss are:

$$\nabla_{\boldsymbol{\omega}} \mathcal{L}_{\mathcal{D}_{\text{test}}}(f_{\boldsymbol{\theta}}) = -\frac{1}{m} \sum_{i=1}^{m} \sum_{k=1}^{K} \nabla_{\boldsymbol{\omega}} \left( \tilde{y}_i'^{(k)} \log s_{\boldsymbol{\omega}}^{(k)}(f_{\mathbf{g}}(\tilde{\mathbf{x}}_i)) \right). \tag{7}$$

Note that our method requires neither gradients of the whole parameter set of the pre-trained model nor iterative training hinting at its high computational efficiency.

**GRDNORM Score.** Now, we can define the GRDNORM Score using a vector norm of gradients of the last layer. The score is expressed as follows:

$$S(\mathcal{D}_{\text{test}}) = \|\nabla_{\boldsymbol{\omega}} \mathcal{L}_{\mathcal{D}_{\text{test}}}(f_{\boldsymbol{\theta}})\|_p, \tag{8}$$

where $\|\cdot\|_p$ denotes $L_p$-norm, and $\boldsymbol{\omega}$ is the classification component of the network of parameters $\boldsymbol{\theta}$.

## 3.2 THEORETICAL ANALYSIS

In this section, we provide some theoretical insights into our method. We first clarify the connection between the true target cross-entropy error and the norm of the gradients. Then, we show that the gradient norm is upper-bounded by a weighted sum of the norm of the inputs.

**Notations.** For the sake of simplicity, we assume the feature extractor $f_{\mathbf{g}}$ is fixed and, by abuse of notation, we use $\mathbf{x}$ instead of $f_{\mathbf{g}}(\mathbf{x})$. Reusing the notations introduced in Section 2, the true target cross-entropy error writes

$$\mathcal{L}_T(\boldsymbol{\omega}) = -\mathbb{E}_{P_T(\mathbf{x},y)} \sum_k y^{(k)} \log s_{\boldsymbol{\omega}}^{(k)}(\mathbf{x}),$$

where $\boldsymbol{\omega} = (\mathbf{w}_k)_{k=1}^{K} \in \mathbb{R}^{D \times K}$ are the parameters of the linear classification layer $f_{\boldsymbol{\omega}}$. For the ease of notation, the gradient of $\mathcal{L}_T$ w.r.t $\boldsymbol{\omega}$ is denoted by $\nabla \mathcal{L}_T$.

The following theorem, whose proof we defer to Appendix H.1, makes the connection between the true risk and the $L_p$-norm of the gradient explicit.

**Theorem 3.1** (Connection between the true risk and the $L_p$-norm of the gradient). *Let* $\mathbf{c} \in \mathbb{R}^{D \times K}$ *and* $\mathbf{c}' \in \mathbb{R}^{D \times K}$ *be two linear classifiers. For any* $p, q \geq 1$ *such that* $\frac{1}{p} + \frac{1}{q} = 1$, *we have that*

$$|\mathcal{L}_T(\mathbf{c}') - \mathcal{L}_T(\mathbf{c})| \leq \max_{\boldsymbol{\omega} \in \{\mathbf{c}', \mathbf{c}\}} (\|\nabla \mathcal{L}_T(\boldsymbol{\omega})\|_p) \cdot \|\mathbf{c}' - \mathbf{c}\|_q.$$

The left-hand side here is the difference in terms of the true risks obtained for the same distribution with two different classifiers. The right-hand side shows how this difference is controlled by the maximum gradient norm over the two classifiers and a term capturing how far the two are apart.

In the context of the proposed approach, we want to know the true risk of the source classifier $\boldsymbol{\omega}_s$ on the OOD data and its change after one step of gradient descent. The following corollary applies Theorem 3.1 to characterize this exact case.

**Corollary 3.2** (Connection after one gradient update). *Let* $\mathbf{c}$ *be the classifier obtained from* $\boldsymbol{\omega}_s$ *after one gradient descent step, i.e.,* $\mathbf{c} = \boldsymbol{\omega}_s - \eta \cdot \nabla \mathcal{L}_T(\boldsymbol{\omega}_s)$ *with* $\eta \geq 0$. *Then, we have that*

$$|\mathcal{L}_T(\boldsymbol{\omega}_s) - \mathcal{L}_T(\mathbf{c})| \leq \eta \cdot \max_{\boldsymbol{\omega} \in \{\boldsymbol{\omega}_s, \mathbf{c}\}} (\|\nabla \mathcal{L}_T(\boldsymbol{\omega})\|_p) \cdot \|\nabla \mathcal{L}_T(\boldsymbol{\omega}_s)\|_q.$$

*Proof.* The proof follows from Theorem 3.1 by noting that $\|\boldsymbol{\omega}_s - \mathbf{c}\|_q = \eta \|\nabla \mathcal{L}_T(\boldsymbol{\omega}_s)\|_q$. $\qquad \square$

Note that in this case $\mathcal{L}_T(\boldsymbol{\omega}_s)$ can be seen as a term providing the true OOD risk after pseudo-labeling it with the source classifier. This shows the importance of pseudo-labeling as it acts as a departure point for obtaining a meaningful estimate of the right-hand side. When the latter is meaningful, the gradient norm on the right-hand side controls it together with a magnitude that tells us how far we went after one step of backpropagation.

In the next theorem, we provide an upper bound on the $L_p$-norm of the gradient as a weighted sum of the $L_p$-norm of the inputs. The proof is deferred in Appendix H.2.

**Theorem 3.3** (Upper-bounding the norm of the gradient). *For any* $p \geq 1$, *the* $L_p$-*norm of the gradient can be upper-bounded as follows:*

$$\|\nabla \mathcal{L}_T(\boldsymbol{\omega})\|_p \leq \mathbb{E}_{P_T(\mathbf{x}, y)} \alpha(\boldsymbol{\omega}, \mathbf{x}, y) \cdot \|\mathbf{x}\|_p, \quad \forall \boldsymbol{\omega} \in \mathbb{R}^{D \times K}$$

*where* $\alpha(\boldsymbol{\omega}, \mathbf{x}, y) = 1 - s_{\boldsymbol{\omega}}^{(k_y)}(\mathbf{x})$, *with* $k_y$ *such that* $y^{(k_y)} = 1$.

Hence, the norm of the gradient is upper-bounded by a weighted combination of the norm of the inputs, where the weight $\alpha(\boldsymbol{\omega}, \mathbf{x}, y) \in [0, 1]$ conveys how well the model predicts on $\mathbf{x}$. In the case of perfect classification, the upper bound is tight and equal to 0. In practice, as we do not have access to the true risk, its gradient can be approximated by the proposed GRDNORM Score and pseudo-label test data by Eq. 5. As we said earlier, this requires the model to be well calibrated, which is conventional to assume for self-training methods (Amini et al., 2023) including the approach of Yu et al. (2022b). Then, the network projects OOD data into the low confidence regions, and the gradient for these examples will be large as we need to update $\boldsymbol{\omega}_s$ significantly to fit them.

## 4 EXPERIMENTS

### 4.1 EXPERIMENTAL SETUP

**Pre-training datasets.** For pre-training the neural network, we use CIFAR-10, CIFAR-100 (Krizhevsky & Hinton, 2009), TinyImageNet (Le & Yang, 2015), ImageNet (Deng et al., 2009), Office-31 (Saenko et al., 2010), Office-Home (Venkateswara et al., 2017), Camelyon17-WILDS (Koh et al., 2021), and BREEDS (Santurkar et al., 2020) which leverages class hierarchy of ImageNet (Deng et al., 2009) to create 4 datasets including Living-17, Nonliving-26, Entity-13 and Entity-30. In particular, to avoid time-consuming training, we directly utilize publicly available models pre-trained on Imagenet. Office-31 (Saenko et al., 2010) is collected from 3 different domains: Amazon, DSLR, and Webcam, while Office-Home (Venkateswara et al., 2017) consists of 4 domains: Artistic, Clip Art, Product and Real-World. For these two datasets, we train a neural network on every domain, which means we have 3 pre-trained models for Office-31 and 4 pre-trained models for Office-Home.

**OOD datasets.** In our comprehensive evaluation, we consider 11 datasets with 3 types of distribution shifts: synthetic, natural, and novel subpopulation shift. To verify the effectiveness of our method under the synthetic shift, we use CIFAR-10C, CIFAR-100C, and ImageNet-C (Hendrycks & Dietterich, 2019) that span 19 types of corruption across 5 severity levels, as well as TinyImageNet-C (Hendrycks & Dietterich, 2019) with 15 types of corruption and 5 severity levels. For the natural shift,

Table 1: Performance comparison on 11 benchmark datasets with ResNet18, ResNet50 and WRN-50-2, where $R^2$ refers to coefficients of determination, and $\rho$ refers to the absolute value of Spearman correlation coefficients (higher is better). The best results are highlighted in **bold**.

| Dataset | Network | Rotation $R^2$ | $\rho$ | ConfScore $R^2$ | $\rho$ | Entropy $R^2$ | $\rho$ | AgreeScore $R^2$ | $\rho$ | ATC $R^2$ | $\rho$ | Fréchet $R^2$ | $\rho$ | Dispersion $R^2$ | $\rho$ | ProjNorm $R^2$ | $\rho$ | Ours $R^2$ | $\rho$ |
|---|---|---|---|---|---|---|---|---|---|---|---|---|---|---|---|---|---|---|---|
| CIFAR 10 | ResNet18 | 0.822 | 0.951 | 0.869 | 0.985 | 0.899 | 0.987 | 0.663 | 0.929 | 0.884 | 0.985 | 0.950 | 0.971 | 0.968 | 0.990 | 0.936 | 0.982 | **0.971** | **0.994** |
| | ResNet50 | 0.835 | 0.961 | 0.935 | 0.993 | 0.945 | **0.994** | 0.835 | 0.985 | 0.946 | **0.994** | 0.858 | 0.964 | **0.987** | 0.990 | 0.944 | 0.989 | 0.969 | 0.993 |
| | WRN-50-2 | 0.862 | 0.976 | 0.943 | **0.994** | 0.942 | **0.994** | 0.856 | 0.986 | 0.947 | **0.994** | 0.814 | 0.973 | 0.962 | 0.988 | 0.961 | 0.989 | **0.971** | **0.994** |
| | Average | 0.840 | 0.963 | 0.916 | 0.991 | 0.930 | 0.992 | 0.785 | 0.967 | 0.926 | 0.991 | 0.874 | 0.970 | **0.972** | 0.990 | 0.947 | 0.987 | 0.970 | **0.994** |
| CIFAR 100 | ResNet18 | 0.860 | 0.936 | 0.916 | 0.985 | 0.891 | 0.979 | 0.902 | 0.973 | 0.938 | 0.986 | 0.888 | 0.968 | 0.952 | 0.988 | 0.979 | 0.980 | **0.987** | **0.996** |
| | ResNet50 | 0.908 | 0.962 | 0.919 | 0.984 | 0.884 | 0.977 | 0.922 | 0.982 | 0.921 | 0.984 | 0.837 | 0.972 | 0.951 | 0.985 | 0.988 | 0.991 | **0.991** | **0.997** |
| | WRN-50-2 | 0.924 | 0.970 | 0.971 | 0.984 | 0.968 | 0.981 | 0.955 | 0.977 | 0.978 | 0.993 | 0.865 | 0.987 | 0.980 | 0.991 | 0.990 | 0.991 | **0.995** | **0.998** |
| | Average | 0.898 | 0.956 | 0.936 | 0.987 | 0.915 | 0.983 | 0.927 | 0.982 | 0.946 | 0.988 | 0.864 | 0.976 | 0.962 | 0.988 | 0.985 | 0.987 | **0.991** | **0.997** |
| TinyImageNet | ResNet18 | 0.786 | 0.946 | 0.670 | 0.869 | 0.592 | 0.842 | 0.561 | 0.853 | 0.751 | 0.945 | 0.826 | 0.970 | 0.966 | 0.986 | 0.970 | 0.981 | **0.971** | **0.994** |
| | ResNet50 | 0.786 | 0.947 | 0.670 | 0.869 | 0.651 | 0.892 | 0.560 | 0.853 | 0.751 | 0.945 | 0.826 | 0.971 | 0.977 | 0.986 | 0.979 | 0.987 | **0.980** | **0.995** |
| | WRNt-50-2 | 0.878 | 0.967 | 0.757 | 0.951 | 0.704 | 0.935 | 0.654 | 0.904 | 0.635 | 0.897 | 0.884 | 0.984 | 0.968 | 0.986 | 0.965 | 0.983 | **0.975** | **0.996** |
| | Average | 0.805 | 0.959 | 0.727 | 0.920 | 0.650 | 0.890 | 0.599 | 0.878 | 0.693 | 0.921 | 0.847 | 0.976 | 0.970 | 0.987 | 0.972 | 0.984 | **0.976** | **0.995** |
| ImageNet | ResNet18 | - | - | 0.979 | 0.991 | 0.963 | 0.991 | - | - | 0.974 | 0.983 | 0.802 | 0.974 | 0.940 | 0.971 | 0.975 | 0.993 | **0.986** | **0.996** |
| | ResNet50 | - | - | 0.980 | 0.994 | 0.967 | 0.992 | - | - | 0.970 | 0.983 | 0.855 | 0.974 | 0.938 | 0.968 | 0.986 | 0.993 | **0.987** | **0.996** |
| | WRNt-50-2 | - | - | 0.983 | 0.991 | 0.963 | 0.991 | - | - | 0.983 | 0.993 | 0.909 | 0.988 | 0.939 | 0.976 | 0.978 | 0.993 | **0.984** | **0.998** |
| | Average | - | - | 0.981 | 0.993 | 0.969 | 0.992 | - | - | 0.976 | 0.987 | 0.855 | 0.979 | 0.939 | 0.972 | 0.980 | 0.993 | **0.986** | **0.997** |
| Office-31 | ResNet18 | **0.753** | **0.942** | 0.470 | 0.828 | 0.322 | 0.714 | 0.003 | 0.085 | 0.843 | 0.942 | 0.143 | 0.257 | 0.618 | 0.714 | 0.099 | 0.428 | 0.675 | 0.829 |
| | ResNet50 | 0.391 | 0.828 | 0.485 | 0.828 | 0.354 | 0.828 | 0.011 | 0.463 | 0.532 | 0.485 | 0.034 | 0.257 | 0.578 | 0.714 | 0.240 | 0.428 | **0.604** | **0.829** |
| | WRN-50-2 | 0.577 | 0.6 | 524 | 0.714 | 0.424 | 0.714 | 0.002 | 0.257 | 0.405 | 0.942 | 0.034 | 0.142 | **0.671** | 0.714 | 0.147 | 0.143 | 0.544 | **0.829** |
| | Average | 0.567 | 0.790 | 0.493 | 0.790 | 0.367 | 0.276 | 0.006 | 0.211 | 0.593 | 0.790 | 0.071 | 0.047 | **0.622** | 0.714 | 0.162 | 0.333 | 0.608 | **0.829** |
| Office-Home | ResNet18 | 0.822 | 0.930 | 0.795 | 0.909 | 0.761 | 0.881 | 0.054 | 0.146 | 0.571 | 0.615 | 0.605 | 0.755 | 0.453 | 0.664 | 0.064 | 0.202 | **0.876** | **0.909** |
| | ResNet50 | **0.851** | **0.944** | 0.769 | 0.895 | 0.742 | 0.853 | 0.026 | 0.216 | 0.487 | 0.734 | 0.607 | 0.685 | 0.383 | 0.727 | 0.169 | 0.475 | 0.829 | 0.944 |
| | WRN-50-2 | **0.823** | **0.958** | 0.741 | 0.874 | 0.696 | 0.846 | 0.132 | 0.405 | 0.383 | 0.643 | 0.589 | 0.706 | 0.456 | 0.713 | 0.172 | 0.531 | 0.809 | 0.916 |
| | Average | 0.832 | 0.944 | 0.768 | 0.892 | 0.733 | 0.860 | 0.071 | 0.256 | 0.480 | 0.664 | 0.601 | 0.715 | 0.431 | 0.702 | 0.135 | 0.403 | **0.837** | **0.923** |
| Camelyon17-WILDS | ResNet18 | 0.944 | **1.000** | 0.980 | **1.000** | 0.980 | **1.000** | 0.977 | **1.000** | 0.981 | **1.000** | 0.988 | **1.000** | 0.992 | **1.000** | 0.612 | 0.500 | **0.996** | **1.000** |
| | ResNet50 | 0.931 | **1.000** | 0.994 | **1.000** | 0.993 | **1.000** | 0.998 | **1.000** | 0.993 | **1.000** | 0.971 | **1.000** | 0.012 | 0.500 | 0.811 | **1.000** | 0.999 | 1.000 |
| | WRN-50-2 | 0.918 | **1.000** | 0.944 | **1.000** | 0.945 | **1.000** | 0.965 | **1.000** | 0.942 | **1.000** | 0.994 | **1.000** | 0.001 | 0.500 | 0.789 | 0.500 | **0.997** | **1.000** |
| | Average | 0.931 | **1.000** | 0.973 | **1.000** | 0.980 | **1.000** | 0.982 | **1.000** | 0.972 | **1.000** | 0.984 | **1.000** | 0.334 | 0.667 | 0.737 | 0.667 | **0.998** | **1.000** |
| Entity-13 | ResNet18 | 0.927 | 0.961 | 0.795 | 0.940 | 0.794 | 0.935 | 0.543 | 0.919 | 0.823 | 0.945 | 0.950 | 0.981 | 0.937 | 0.968 | 0.952 | 0.981 | **0.969** | **0.991** |
| | ResNet50 | 0.932 | 0.976 | 0.728 | 0.941 | 0.698 | 0.928 | 0.901 | 0.964 | 0.783 | 0.950 | 0.903 | 0.959 | 0.764 | 0.892 | 0.944 | 0.974 | **0.960** | **0.995** |
| | WRN-50-2 | 0.939 | 0.983 | 0.930 | 0.977 | 0.919 | 0.973 | 0.871 | 0.935 | 0.936 | 0.980 | 0.906 | 0.958 | 0.815 | 0.905 | 0.950 | 0.977 | **0.968** | **0.995** |
| | Average | 0.933 | 0.973 | 0.817 | 0.953 | 0.804 | 0.945 | 772 | 0.939 | 0.847 | 0.958 | 0.920 | 0.966 | 0.948 | 0.977 | 0.839 | 0.922 | **0.966** | **0.994** |
| Entity-30 | ResNet18 | 0.964 | 0.979 | 0.570 | 0.836 | 0.553 | 0.832 | 0.542 | 0.935 | 0.611 | 0.845 | 0.849 | 0.978 | 0.929 | 0.968 | 0.952 | 0.987 | **0.970** | **0.995** |
| | ResNet50 | **0.961** | 0.980 | 0.878 | 0.969 | 0.838 | 0.956 | 0.914 | 0.975 | 0.924 | 0.973 | 0.835 | 0.956 | 0.783 | 0.914 | 0.937 | 0.986 | 0.957 | **0.996** |
| | WRN-50-2 | 0.940 | 0.978 | 0.897 | 0.974 | 0.878 | 0.970 | 0.826 | 0.955 | 0.936 | 0.954 | 0.927 | 0.973 | 0.927 | 0.973 | 0.986 | 0.949 | **0.959** | 0.994 |
| | Average | 0.955 | 0.978 | 0.781 | 0.926 | 0.756 | 0.919 | 0.728 | 0.956 | 0.823 | 0.934 | 0.871 | 0.969 | 0.880 | 0.952 | 0.949 | 0.987 | **0.959** | **0.995** |
| Living-17 | ResNet18 | 0.876 | 0.973 | 0.913 | 0.973 | 0.898 | 0.970 | 0.586 | 0.736 | 0.940 | 0.973 | 0.768 | 0.950 | 0.900 | 0.958 | 0.923 | 0.970 | **0.949** | **0.983** |
| | ResNet50 | 0.906 | 0.956 | 0.880 | 0.967 | 0.853 | 0.961 | 0.633 | 0.802 | **0.938** | **0.976** | 0.771 | 0.926 | 0.851 | 0.929 | 0.903 | 0.924 | 0.931 | 0.975 |
| | WRN-50-2 | 0.909 | 0.957 | 0.928 | 0.980 | 0.921 | 0.977 | 0.652 | 0.793 | **0.966** | **0.984** | 0.931 | 0.967 | 0.931 | 0.966 | 0.915 | 0.970 | 0.910 | 0.976 |
| | Average | 0.933 | 0.974 | 0.907 | 0.973 | 0.814 | 0.969 | 0.623 | 0.777 | **0.948** | **0.978** | 0.817 | 0.949 | 0.894 | 0.951 | 0.913 | 0.969 | 0.930 | **0.978** |
| Nonliving-26 | ResNet18 | 0.906 | 0.955 | 0.781 | 0.925 | 0.739 | 0.909 | 0.543 | 0.810 | 0.854 | 0.939 | 0.914 | 0.980 | **0.958** | 0.981 | 0.939 | 0.978 | 0.953 | **0.983** |
| | ResNet50 | 0.916 | 0.970 | 0.832 | 0.942 | 0.776 | 0.918 | 0.638 | 0.837 | 0.893 | 0.960 | 0.848 | 0.950 | 0.805 | 0.907 | 0.873 | 0.972 | **0.945** | **0.989** |
| | WRN-50-2 | 0.917 | 0.977 | 0.932 | 0.971 | 0.912 | 0.959 | 0.676 | 0.861 | **0.945** | 0.969 | 0.885 | 0.942 | 0.893 | 0.939 | 0.924 | 0.973 | 0.937 | **0.985** |
| | Average | 0.913 | 0.967 | 0.849 | 0.946 | 0.809 | 0.929 | 0.618 | 0.836 | 0.897 | 0.956 | 0.882 | 0.957 | 0.913 | 0.974 | 0.886 | 0.943 | **0.945** | **0.985** |

we use the domains excluded from training from Office-31, Office-Home, and Camelyon17-WILDS as the OOD datasets. For the novel subpopulation shift, we consider the BREEDS benchmark which Living-17, Nonliving-26, Entity-13 and Entity-30 are constructed from ImageNet-C.

**Training details.** To show the versatility of our approach across different architectures, we perform all our experiments on ResNet18, ResNet50 (He et al., 2016) and WRN-50-2 (Zagoruyko & Komodakis, 2016) models. We train them for 20 epochs for CIFAR-10 (Krizhevsky & Hinton, 2009) and for 50 epochs for the other datasets. In all cases, we use SGD with a learning rate of $10^{-3}$, cosine learning rate decay (Loshchilov & Hutter, 2016), a momentum of 0.9, and a batch size of 128.

**Evaluation metrics.** We measure the performance of all competing methods using the coefficients of determination ($R^2$) and Spearman correlation coefficients ($\rho$) calculated between the baseline scores and the true OOD error. To compare the computational efficiency with two self-training methods, we calculate the average evaluation time needed for every test dataset.

**Baseline methods.** We consider 8 baselines commonly evaluated in the OOD estimation error studies, notably: *Rotation Prediction* (Rotation) (Deng et al., 2021), *Averaged Confidence* (ConfScore)

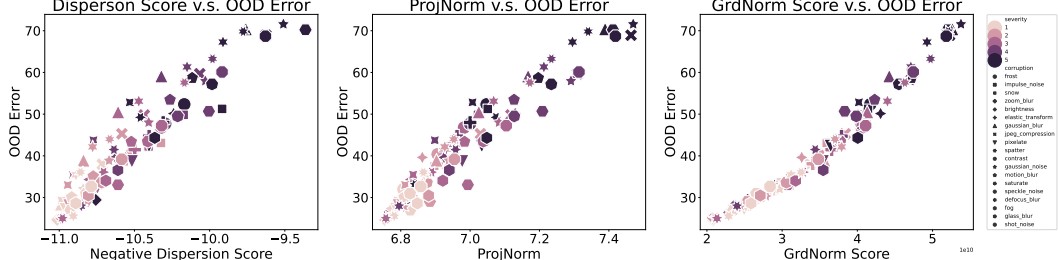

Figure 2: OOD error prediction versus True OOD error on Entity-13 with ResNet18. We compare the performance of GRDNORM Score with that of Dispersion Score and ProjNorm via scatter plots. Each point represents one dataset under certain corruption and certain severity, where different shapes represent different types of corruption, and darker color represents the higher severity level.

(Hendrycks & Gimpel, 2016), *Entropy* (Guillory et al., 2021), *Agreement Score* (AgreeScore) (Jiang et al., 2021), *Averaged Threshold Confidence* (ATC) (Garg et al., 2022), *AutoEval* (Fréchet) (Deng & Zheng, 2021), *Dispersion Score* (Dispersion) (Xie et al., 2023) and *ProjNorm* (Yu et al., 2022b), where the first seven methods are training-free and the last method is an instance of self-training approaches. In general, Rotation and Agreescore address the problem by constructing an unsupervised loss. ConfScore, Entropy, and ATC estimate the OOD error using the model predictions. Fréchet and ProjNorm gauge the distribution discrepancy as the OOD error estimation from the feature-representation level and the parameter level, and Dispersion measures the separability of the test feature representation. More details are introduced in Appendix C.

## 4.2 MAIN TAKE-AWAYS

**GRDNORM correlates with OOD error stronger than baselines.** In Table 1, we present the OOD error estimation performance on 11 benchmark datasets across 3 model architectures as measured by $R^2$ and $\rho$. We observe that GRDNORM outperforms existing methods under diverse distribution shifts. Our method achieves an average $R^2$ higher than 0.99 on CIFAR-100, while the average $R^2$ of the other baseline methods is all below that value. In addition, our method performs stably across different distribution shifts compared with the other existing algorithms. For example, Rotation performs well under the natural shift but experiences a dramatic performance drop under the synthetic shift, ranking from the second best to the eighth. However, our method achieves consistently high performance, ranking the best on average across the three types of distribution shifts.

Furthermore, we also provide the visualization of estimation performance in Fig. 2, where we present the scatter plots for Dispersion Score, ProjNorm and GRDNORM Score on Entity-13 with ResNet18. From this figure, we find that GRDNORM Score and OOD error perform a strong linear relationship, while the other state-of-the-art methods fail especially when the OOD error is high. This phenomenon clearly demonstrates the superiority of GRDNORM Score in OOD error estimation. In the next paragraph, we also demonstrate the computational efficiency of our approach.

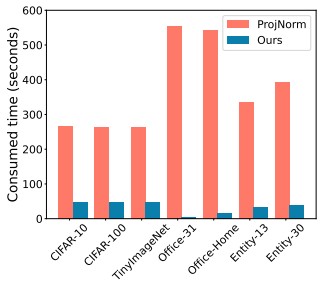

Figure 1: Runtime comparison of two self-training approaches with ResNet50.

**GRDNORM is a more efficient self-training approach.** In the list of baselines, both our method and ProjNorm (Yu et al., 2022b) belong to self-training methods (Amini et al., 2023). The latter, however, requires costly iterative training on the neural network during evaluation to obtain the complete set of fine-tuned parameters to calculate the distribution discrepancy in network parameters. Compared with ProjNorm, our method only trains the model for one epoch and collects the gradients of the linear classification layer to calculate the gradient norm, which is much more computationally efficient. Fig. 1 presents the comparison of computational efficiency between the two methods on 7 datasets with ResNet50. From this figure, we can see that our method is up to 80% faster than ProjNorm on average. This difference is striking on Office-31 dataset, where our method is not only two orders of magnitude faster than ProjNorm but also improves the $R^2$ score by a factor of 2.

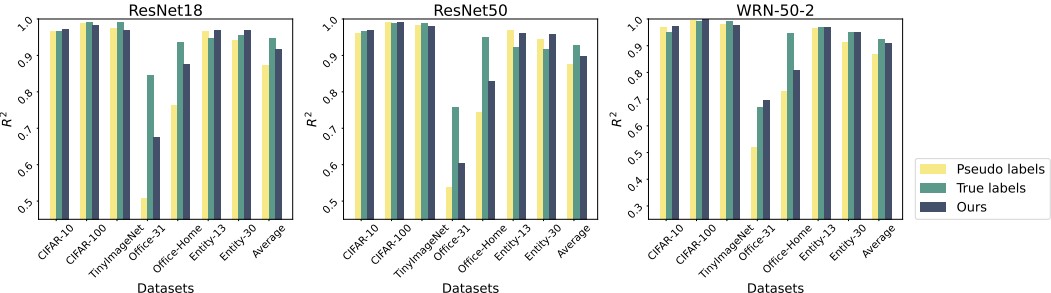

Figure 3: Performance comparison on different label generation strategies including our label generation strategy (i.e., *Ours*), using pseudo labels only (i.e., *Pseudo labels*), and using true labels only (i.e., *True labels*) with ResNet18, ResNet50 and WRN-50-2.

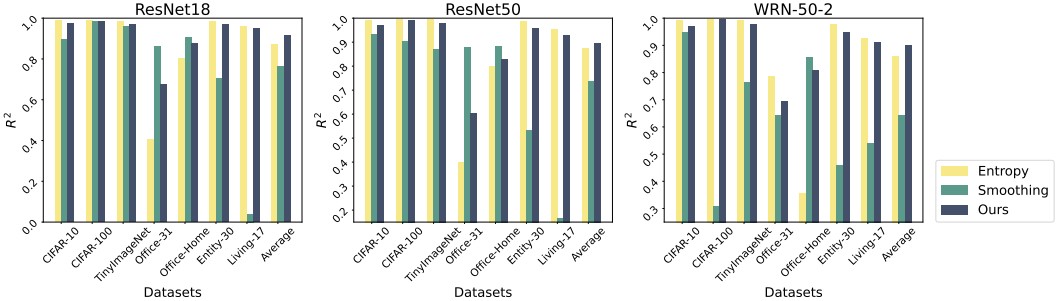

Figure 4: Performance comparison on different types of losses including the cross-entropy loss (i.e., *ours*), the entropy loss for samples with low confidence (i.e., *Entropy*), and the cross-entropy loss with label smoothing (i.e., *Smoothing*) with ResNet18, ResNet50, and WRN-50-2.

## 5 ABLATION STUDY

Our method involves several algorithmic choices such as the proposed pseudo-labeling strategy, the loss used for backpropagation as well as the number of gradient steps, and the $p$ values of the norm used. Below, we provide an ablation study related to all these choices.

**Random pseudo-labels boost the performance under natural shift.** In Fig. 3, we conduct an ablation study to verify the effectiveness of our label generation strategy by comparing it with ground-truth labels and with full pseudo-labeling. From the figure, we observe that while comparable with the ground truth under the synthetic drift and the subpopulation shift, pseudo-labels lead to a drastic drop in performance under the natural shift. This phenomenon is possibly caused by imprecise gradient calculation based on incorrect pseudo labels. Our labeling strategy performs better on average suggesting that random labels provide certain robustness of the score under natural shift.

**Cross-entropy loss is robust to different shifts.** To demonstrate the impact of different losses on the OOD estimation performance, we compare the standard cross-entropy loss used in our method with the entropy loss for samples with low confidence. Its detailed introduction is in Appendix D. On the other hand, since label smoothing is a simple yet efficient tool for classification performance improvement and model calibration (Müller et al., 2019), we also conduct an ablation study to verify the effectiveness of this regularization by setting the smoothing rate as 0.4.

Fig. 4 illustrates this comparison revealing that standard cross-entropy loss is the most robust choice across different types of distribution shifts. We also note that the entropy loss enhances the estimation performance under the synthetic and the novel subpopulation shifts, but struggles under the natural shift. On the contrary, the label smoothing regularization can increase the performance under the natural shift but decreases it under the synthetic and the novel subpopulation shift.

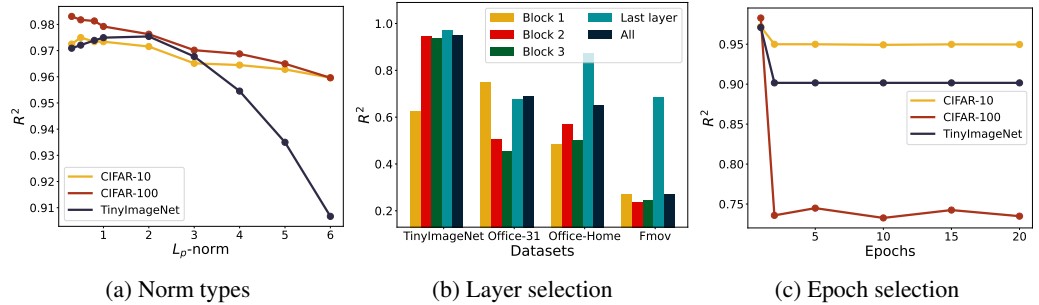

|                        |                        |                        |
| :--------------------: | :--------------------: | :--------------------: |
| (a) Norm types         | (b) Layer selection    | (c) Epoch selection    |

Figure 5: Sensitivity analysis on the effect of (a) norm types, (b) layer selection for gradients, and (c) epoch selection. The first and the third experiments are conducted on CIFAR-10C, CIFAR-100C, and TinyImageNet-C, while the second experiment includes TinyImageNet-C, Office-31, Office-Home, and WILDS-FMoV (Koh et al., 2021). All experiments are conducted with ResNet18.

**Choosing smaller $p$ for better estimation performance.** To illustrate the effect of the norm choice (i.e., $p$ in $L_p$-norm) on OOD error estimation, we conduct a sensitivity analysis on three datasets with ResNet18 and gain Fig. 5a. From this figure, we find that there is an obvious decreasing trend as $p$ becomes larger. Especially, when $p$ is smaller than 1, the estimation performance is able to fluctuate within a satisfying range. This is probably because a smaller $p$ (i.e., $0 < p < 1$) makes the $L_p$-norm be more suitable for the high-dimensional space, while the $L_p$-norm with $p \geq 1$ is likely to ignore gradients that are close to 0 (Wang et al., 2016; Huang et al., 2021). So in this paper, we set $p$ as 0.3 for all datasets and model structures. The following remark provides some theoretical insights into the $L_p$-norm of the gradient for $0 < p < 1$. The proof is deferred in Appendix H.3

**Remark 5.1** (Case $0 < p < 1$). *Let* $\mathbf{c}$ *be the classifier obtained from* $\boldsymbol{\omega}_s$ *after one gradient descent step, i.e.,* $\mathbf{c} = \boldsymbol{\omega}_s - \eta \cdot \nabla \mathcal{L}_T(\boldsymbol{\omega}_s)$ *with* $\eta \geq 0$. *For any* $p \in (0, 1)$, *we have that*

$$\eta \|\nabla \mathcal{L}_T(\boldsymbol{\omega}_s)\|_p \leq |\|\mathbf{c}\|_p - \|\boldsymbol{\omega}_s\|_p|.$$

**Gradients from the last layer can provide sufficient information.** In this part, we aim to understand whether backpropagating through other layers in the neural network can provide a better OOD error estimation. For this, we separate the feature extractor $f_{\mathbf{g}}$ into 3 blocks of layers with roughly equal size and calculate the GRDNORM scores for each of them. Additionally, we also try to gather the gradients over the whole network. Fig. 5b plots the obtained results on 4 datasets, suggesting that the last layer provides sufficient information to predict the true OOD error.

**No gains after 1 epoch of backpropagation.** Here, we train the neural network for $r$ epochs, where $r \in \{1, 2, 5, 10, 15, 20\}$, and store the gradient vectors of the classification layer for each value of $r$. Fig. 5c suggests that the gradient norms after 1 step are sufficient to predict the model performance under distribution shifts. Further training gradually degrades the performance with the increasing $r$. The reason behind the phenomenon is that the gradients in the first epoch contain the most abundant information about the training dataset, while the neural network fine-tuned on the test dataset for several epochs is going to forget previous training categories (Kemker et al., 2018).

## 6 CONCLUSION

In this paper, we demonstrated a strong linear relationship between the magnitude of gradients and the model performance under distribution shifts. We proposed GRDNORM Score, a simple yet efficient method to estimate the ground-truth OOD error by measuring the gradient norm of the last classification layer. Our method can achieve consistently better performance across different distribution shifts than previous works with high computational efficiency. Furthermore, it does not require the detailed architecture of the feature extractors and training datasets. Those properties guarantee that our method can be easily deployed in the real world, and meet practical demands, such as large models and confidential information. We hope that our research throws light on the future exploration of gradient norm for OOD error estimation.

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

## A    PSEUDO CODE OF GRDNORM SCORE

Our proposed GRDNORM Score for OOD error estimation can be calculated as shown in Algorithm 1.

---

**Algorithm 1** OOD Error Estimation via GRDNORM Score

---

**Input:** OOD test dataset $\tilde{\mathcal{D}} = \{\tilde{\mathbf{x}}_i\}_{i=1}^m$, a pre-trained model $f_{\boldsymbol{\theta}} = f_{\mathbf{g}} \circ f_{\boldsymbol{\omega}}$ (feature extractor $f_{\mathbf{g}}$ and classifier $f_{\boldsymbol{\omega}}$), a threshold value $\tau$.
**Output:** The GRDNORM Score.
**for** each OOD instance $\tilde{\mathbf{x}}_i$ **do**
    Obtain the maximum softmax probability via $\tilde{r}_i = \max_k s_{\boldsymbol{\omega}}^{(k)}(f_{\mathbf{g}}(\tilde{\mathbf{x}}_i))$ .
    **if** $\tilde{r}_i > \tau$ **then**
        Obtain pseudo labels via $\tilde{y}_i' = \arg\max_k f_{\boldsymbol{\theta}}(\tilde{\mathbf{x}}_i)$,
    **else**
        Obtain random labels via $\tilde{y}_i' \sim U[1, K]$.
    **end if**
**end for**
Calculate the cross-entropy loss using assigned labels $\tilde{y}_i$ via Eq. 6.
Calculate gradients of the weights in the classification layer via Eq. 7.
Calculate GRDNORM Score $S(\tilde{\mathcal{D}})$ via Eq. 8.

---

## B    RELATED WORK

In this section, we first introduce the relevant literature in the field of our interest, OOD error estimation. Then, we list two fields, OOD detection and generalization error bounds, with their respective related work to clarify the difference from OOD error estimation, avoiding potential misunderstanding and confusion with our problem.

**OOD error estimation.** OOD error estimation is a vital topic in practical applications due to frequent distribution shifts and infeasible ground-truth labels of the test samples. To comprehensively understand this field, we introduce two main existing settings which are related to this topic.

1. Some works aim to estimate the test error or gauge the accuracy discrepancy between the training and the test set only via the training data (Corneanu et al., 2020; Jiang et al., 2019; Neyshabur et al., 2017; Unterthiner et al., 2020; Yak et al., 2019; Martin & Mahoney, 2020). For example, the model-architecture-based algorithm (Corneanu et al., 2020) derives plenty of persistent topology properties from the training data, which can identify when the model learns to generalize to unseen datasets. However, those algorithms are deployed under the assumption that the training and the test data are drawn from the same distribution, which means they are vulnerable to distribution shifts.

2. Our work belongs to the second setting, which aims to estimate the classification accuracy of a specific OOD test dataset during evaluation using unlabeled test samples and/or labeled training datasets. The main research direction is to explore the negative relationship between the distribution discrepancy and model performance from the space of features (Deng & Zheng, 2021), parameters (Yu et al., 2022b) and labels (Lu et al., 2023). Another popular direction is to design an OOD estimation score via the softmax outputs of the test samples (Guillory et al., 2021; Jiang et al., 2021; Guillory et al., 2021; Garg et al., 2022), which heavily relies on model calibration. Some works also learn from the field of unsupervised learning, such as agreement across multiple classifiers (Jiang et al., 2021; Madani et al., 2004; Platanios et al., 2016; 2017) and image rotation (Deng et al., 2021). In addition, the property of the test datasets presented during evaluation has been also studied recently (Xie et al., 2023). To the best of our knowledge, our work is the first to study the linear relationship between the gradients and model performance.

**OOD detection.** Out-of-distribution (OOD) detection is another essential building block for machine learning safety, whose goal is to determine whether a given sample is in-distribution (ID) or out-of-distribution (Hendrycks & Gimpel, 2016; Hendrycks et al., 2018; Liu et al., 2020; Yang et al., 2021;

Liang et al., 2017). For example, a common baseline uses the maximum softmax probabilities to detect OOD data, assuming that samples with lower softmax probabilities tend to be OOD samples (Hendrycks & Gimpel, 2016). Particularly, gradient norm is also explored in this field under the intuition that ID data need a higher magnitude of gradients than OOD data for adjusting from the softmax probabilities to a uniform distribution (Huang et al., 2021). Nevertheless, OOD detection discusses label-space shifts, where ID and OOD data have disjoint label sets. In contrast, distribution shifts in OOD error estimation do not change the label space shared by the training and the test datasets.

**Remark B.1** (Main differences between OOD detection and OOD error estimation). *The OOD error estimation problem should not be confused with the OOD detection problem which received a significant amount of attention in the literature as well. Indeed, the latter, as considered for instance in Huang et al. (2021); Igoe et al. (2022), seeks to solve a binary classification problem of predicting whether a given test instance is OOD. Additionally, the meaning of OOD in such a setting commonly refers to a dataset with a non-overlapping label set. OOD error estimation requires a dataset-based score, mainly considered in distribution shift cases where the label sets can be overlapping. The two are also evaluated differently: AUROC score for OOD detection and correlation coefficients for error estimation. Those differences are summarized in Table 2. In Appendix E, we show that a recent gradient-based OOD detection method (Huang et al., 2021) is inferior to our approach for OOD error estimation suggesting that the two problems cannot be tackled with the same tools.*

Table 2: Main differences between OOD detection and OOD error estimation.

| Learning Problem | Goal | Scope | Metric |
|---|---|---|---|
| OOD detection | Predict ID/OOD | $\tilde{\mathbf{x}}_i$ | AUROC |
| OOD error estimation | Proxy to test error | $\mathcal{D}_{\text{test}}$ | $R^2$ and $\rho$ |

**Generalization error bounds.** Generalization error, also known as the out-of-sample error, gauges the generalization performance of the hypothesis learned from the training data and applied to previous unseen samples (Hardt et al., 2016; London, 2017; Rivasplata et al., 2018). Many studies try to provide a tight upper bound for generalization error from the view of gradient descent theoretically, which indicates that gradients correlate with the discrepancy between the empirical loss and the population loss (Li et al., 2019; Chatterjee, 2020; Negrea et al., 2019; An et al., 2020). However, those works mainly focus on the generalization performance from seen data to unseen data under the identical distribution, while OOD error estimation discusses a more complex and realistic issue that seen and unseen data come from different distributions.

## C  BASELINE METHODS

**Rotation.**  (Deng et al., 2021) By rotating images from both the training and the test sets with different angles, we can obtain new inputs and their corresponding labels $y_i^r$ which indicate by how many degrees they rotate. During pre-training, an additional classifier about rotation degrees should be learned. Then the *Rotation Prediction* (Rotation) metric can be calculated as:

$$S_r(\mathcal{D}_{\text{test}}) = \frac{1}{m} \sum_{i=1}^{m} \left(\frac{1}{4} \sum_{r \in \{0°, 90°, 180°, 270°\}} (\mathbb{1}(\hat{y}_i^r \neq y_i^r))\right),$$

where $\hat{y}_i^r$ denotes the predicted labels about rotation degrees.

**ConfScore.**  (Hendrycks & Gimpel, 2016) This metric directly leverages the average maximum softmax probability as the estimation of OOD error, which is expressed as:

$$S_{cf}(\mathcal{D}_{\text{test}}) = \frac{1}{m} \sum_{i=1}^{m} \max(\mathrm{s}_{\boldsymbol{\omega}}(f_{\boldsymbol{g}}(\tilde{\mathbf{x}}_i))).$$

**Entropy.**  (Guillory et al., 2021) This metric estimates OOD error via the average entropy loss:

$$S_e(\mathcal{D}_{\text{test}}) = \frac{1}{m} \sum_{i=1}^{m} \sum_{k=1}^{K} \mathrm{s}_{\boldsymbol{\omega}}^{(k)}(f_{\boldsymbol{g}}(\tilde{\mathbf{x}}_i)) \log \mathrm{s}_{\boldsymbol{\omega}}^{(k)}(f_{\boldsymbol{g}}(\tilde{\mathbf{x}}_i)).$$

**AgreeScore.** (Jiang et al., 2021) This method trains two independent neural networks simultaneously during pre-training, and estimates OOD error via the rate of disagreement across the two models:

$$S_{ag}(\mathcal{D}_{\text{test}}) = \frac{1}{m}\sum_{i=1}^{m}\mathbb{1}(\tilde{y}'_{1,i} \neq \tilde{y}'_{2,i}),$$

where $\tilde{y}'_{1,i}$ and $\tilde{y}'_{2,i}$ denote the predicted labels by the two models respectively.

**ATC.** (Garg et al., 2022) It measures how many test samples have a confidence larger than a threshold that is learned from the source distribution. It can be expressed as:

$$S_{atc}(\mathcal{D}_{\text{test}}) = \frac{1}{m}\sum_{i=1}^{m}\mathbb{1}(\sum_{k=1}^{K}\mathrm{s}_{\boldsymbol{\omega}}^{(k)}(f_{\boldsymbol{g}}(\tilde{\mathbf{x}}_i))\log\mathrm{s}_{\boldsymbol{\omega}}^{(k)}(f_{\boldsymbol{g}}(\tilde{\mathbf{x}}_i)) < t),$$

where $t$ is the threshold value learned from the validation set of the training dataset.

**Fréchet.** (Deng & Zheng, 2021) This method utilizes Fréchet distance to measure the distribution gap between the training and the test datasets, which provides the OOD error estimation:

$$S_{fr}(\mathcal{D}_{\text{test}}) = ||\mu_{train} - \mu_{test}|| + Tr(\Sigma_{train} + \Sigma_{test} - 2(\Sigma_{train}\Sigma_{test})^{\frac{1}{2}}),$$

where $\mu_{train}$ and $\mu_{test}$ denote the mean feature vector of $\mathcal{D}$ and $\mathcal{D}_{test}$, respectively. $\Sigma_{train}$ and $\Sigma_{test}$ refer to the covariance matrices of corresponding datasets.

**Dispersion.** (Xie et al., 2023) This paper estimates OOD error by gauging the feature separability of the test dataset in the feature space:

$$S_{dis}(\mathcal{D}_{\text{test}}) = \log\frac{\sum_{k=1}^{K}m_k \cdot ||\bar{\boldsymbol{\mu}} - \tilde{\boldsymbol{\mu}}_k||_2^2}{K-1},$$

where $\boldsymbol{\mu}$ denotes the center of the whole features, and $\boldsymbol{\mu}_k$ denotes the mean of $k^{th}$-class features.

**ProjNorm.** (Yu et al., 2022b) This method fine-tunes the pre-trained model on the test dataset with pseudo-labels, and measures the distribution discrepancy between the training and the test datasets in the parameter level:

$$S_{pro}(\mathcal{D}_{\text{test}}) = ||\tilde{\boldsymbol{\theta}}_{ref} - \tilde{\boldsymbol{\theta}}||_2,$$

where $\boldsymbol{\theta}_{ref}$ denotes the parameters of the pre-trained model, while $\boldsymbol{\theta}$ denotes the parameters of the fine-tuned model.

Those algorithms mentioned in this paper can be summarized as Table 3.

## D  ENTROPY LOSS FOR LOW-CONFIDENCE SAMPLES

In the ablation study, we explore the impact of loss selection on the performance of OOD error estimation. In particular, the detail about the entropy loss for samples with low confidence is expressed as follows: In particular, the entropy loss can be expressed as follows:

$$\mathcal{L}(f_{\boldsymbol{\theta}}(\tilde{\mathbf{x}})) = -\frac{1}{m_1}\sum_{i=1}^{m_1}\sum_{k=1}^{K}\tilde{y}_{i,con>\tau}^{(k)}\log\mathrm{s}_{\boldsymbol{\omega}}^{(k)}(f_{\boldsymbol{g}}(\tilde{x}_i^{con>\tau}))$$

$$-\frac{1}{m_2}\sum_{i=1}^{m_2}\sum_{k=1}^{K}\mathrm{s}_{\boldsymbol{\omega}}^{(k)}(f_{\boldsymbol{g}}(\tilde{\mathbf{x}}_i^{con\leq\tau}))\log\mathrm{s}_{\boldsymbol{\omega}}^{(k)}(f_{\boldsymbol{g}}(\tilde{\mathbf{x}}_i^{con\leq\tau})),$$

where the first term denotes the cross-entropy loss calculated for samples with confidence larger than the threshold value $\tau$, the second term denotes the entropy loss for samples with lower confidence than $\tau$, and $con$ means the sample confidence, $m_1$ and $m_2$ denote the total number of samples with higher confidence and lower confidence than $\tau$, respectively.

Table 3: Method property summary including whether this method belongs to self-training or training-free approaches, and if this method requires training data or specific model architectures.

| Method | Self-training | Training-free | Training-data-free | Architecture-requirement-free |
|--------|:-------------:|:-------------:|:------------------:|:-----------------------------:|
| Rotation | ✗ | ✓ | ✓ | ✗ |
| ConfScore | ✗ | ✓ | ✓ | ✓ |
| Entropy | ✗ | ✓ | ✓ | ✓ |
| Agreement | ✗ | ✓ | ✓ | ✗ |
| ATC | ✗ | ✓ | ✗ | ✓ |
| Fréchet | ✗ | ✓ | ✗ | ✓ |
| Dispersion | ✗ | ✓ | ✓ | ✓ |
| ProjNorm | ✓ | ✗ | ✓ | ✓ |
| Ours | ✓ | ✗ | ✓ | ✓ |

## E    DISCUSSION: RELATION TO HUANG ET AL. (2021)

A current work, GradNorm (Huang et al., 2021), employs gradients to detect OOD samples which labels belong to a different label space from the training data. It gauges the magnitude of gradients in the classification layer, backpropagated from a KL-divergence between the softmax probability and a uniform distribution. Compared with GradNorm, our method bears three critical differences, in terms of the problem setting, methodology, and theoretical insights. We also empirically demonstrate the superiority of our method in Table 4.

1) *Problem setting*: GradNorm focuses on OOD detection, where the label spaces of OOD data and training data are disjoint, while our method aims to estimate the test error without ground-truth test labels, where the training and the OOD label spaces are shared.

2) *Methodology*: Essentially, GradNorm measures the magnitude of gradients from the prediction probability to the uniform distribution, while our method measures it from the source distribution to the target distribution. This basic difference is due to different aims, and is specifically reflected in the design of losses, label generalization strategies and evaluation approaches.

3) *Theoretical insights*: Theoretically, GradNorm captures the joint information between features and outputs to detect OOD data from the oncoming dataset, while our method provides two types of parameter discrepancy information that are beneficial to predicting OOD performance. Formally, we also demonstrate that our score formulates the upper bound of the true OOD error, which further explains the effectiveness of our method.

In Table 4, we present the performance comparison of the two methods in OOD error estimation on 7 datasets across 3 types of distribution shifts with ResNet18. This table illustrates that our method is superior to Huang et al. (2021): for example, our method outperforms Huang et al. (2021) on TinyImageNet with a large margin from 0.894 to 0.971. This shows that comparing the softmax outputs to uniform distribution as done by Huang et al. (2021) is relevant for detecting test samples from a different label space only. However, for OOD error estimation with overlapping labels, estimating the target distribution through pseudo-labeling – rather than assuming it to be uniform – is more informative and achieves much better results.

Table 4: Performance comparison between Huang et al. (2021) and our methods on 7 datasets with ResNet18. The metric used in this table is the coefficient of determination $R^2$. The best results are highlighted in **bold**.

| Method | CIFAR 10 | CIFAR 100 | TinyImageNet | Office-31 | Office-Home | Entity-30 | Living-17 |
|--------|:--------:|:---------:|:------------:|:---------:|:-----------:|:---------:|:---------:|
| (Huang et al., 2021) | 0.951 | 0.978 | 0.894 | 0.596 | 0.848 | 0.964 | 0.942 |
| Ours | **0.972** | **0.983** | **0.971** | **0.675** | **0.876** | **0.970** | **0.949** |

## F    CHOICE OF PROPER THRESHOLD $\tau$

In our experiments (see Section 4), we set the value of $\tau$ as 0.5 across all datasets and network architectures. This choice of $\tau$ is due to the intuition that if a label contains a softmax probability below 0.5, it means that this predicted label has over $50\%$ chances of being wrong. It means that this label has a higher probability of being incorrect than to be correct. Thus, we tend to regard it as an incorrect prediction. To demonstrate the impact of threshold $\tau$ on the final performance, we conduct an ablation study on CIFAR-10 with ResNet18 with varying values of $\tau$. We display in Table 5 the corresponding values of $R^2$. We can observe that the final performance improves and achieves its best value for $\tau$ is 0.5, before decreasing slightly.

Table 5: Performance on CIFAR-10 with ResNet18 for varying value of $\tau$. The metric used in this table is the coefficient of determination $R^2$. The best result is highlighted in **bold**.

| Threshold | 0.0 | 0.1 | 0.2 | 0.3 | 0.4 | 0.5 | 0.6 | 0.7 | 0.8 | 0.9 |
|---|---|---|---|---|---|---|---|---|---|---|
| $R^2$ | 0.963 | 0.963 | 0.964 | 0.965 | 0.971 | **0.972** | 0.967 | 0.962 | 0.963 | 0.959 |

## G    INFLUENCE OF CALIBRATION ERROR

We have mentioned earlier that, in theory, the proposed pseudo-labeling strategy depends on how well the prediction probabilities are calibrated. In degraded cases, this can have a negative impact on our approach, e.g., one can imagine a model that outputs only one-hot probabilities with not a high accuracy. However, this it is generally not the case. Indeed, in practice, we do not need to have a perfectly calibrated model as we employ a mixed strategy that assigns pseudo-labels to high-confidence examples and random labels to low-confidence ones. The recent success of applying self-training models to different problems (Sohn et al., 2020; Dong et al., 2021; Yu et al., 2022a) provides evidence of the suitability of the label generation strategy we adopted.

When we speak of deep neural networks, which are widely accepted to be poorly calibrated, Minderer et al. (2021) showed that modern SOTA image models tend to be well-calibrated across distribution shifts. To demonstrate it empirically, in Table 6, we provide the expected calibration error (ECE, Guo et al. (2017)) of ResNet18, one of the considered base models, depending on a difficulty of test data. For this, we test first on CIFAR-10 (ID), and then on CIFAR-10C corrupted by brightness across diverse severity from 1 to 5. We can see that ECE is very low for ID data and remains relatively low across all levels of corruption severity, which shows that ResNet is quite well-calibrated on CIFAR-10.

Table 6: Expected Error Calibration (ECE) of ResNet18 on CIFAR-10 (ID) and CIFAR-10C corrupted by brightness across diverse severity from 1 to 5.

| Corruption Severity | ID | 1 | 2 | 3 | 4 | 5 |
|---|---|---|---|---|---|---|
| ECE | 0.0067 | 0.0223 | 0.0230 | 0.0243 | 0.0255 | 0.0339 |

On the other hand, in the case of more complex distribution shift like Office-31 data set, we can see that the calibration error has been increased noticeably (Table 7). It is interesting to analyze this result together with Figure 3 of the main paper, where we compared the results between the usual pseudo-labeling strategy and the proposed one. Although our method has room for improvement compared to the oracle method, it is also significantly better than "pseudo-labels", indicating that the proposed label generation strategy is less sensitive to the calibration error.

Table 7: Expected Error Calibration (ECE) of ResNet18 on Office-31 data set.

| Domain | DSLR (ID) | Amazon | Webcam |
|---|---|---|---|
| ECE | 0.2183 | 0.2167 | 0.4408 |

# H    PROOFS

## H.1    PROOF OF THEOREM 3.1

We start by proving the following lemma.

**Lemma H.1.** *For any convex function $f : \mathbb{R}^D \to \mathbb{R}$ and any $p, q \geq 1$ such that $\frac{1}{p} + \frac{1}{q} = 1$, we have:*

$$\forall \mathbf{a}, \mathbf{b} \in dom(f), \quad |f(\mathbf{a}) - f(\mathbf{b})| \leq \max_{\mathbf{c} \in \{\mathbf{a}, \mathbf{b}\}} \{\|\nabla f(\mathbf{c})\|_p\} \cdot \|\mathbf{a} - \mathbf{b}\|_q.$$

*Proof.* Using the fact that $f$ is convex, we have:

$$
\begin{aligned}
f(\mathbf{a}) - f(\mathbf{b}) &\leq \langle \nabla f(\mathbf{a}), \mathbf{a} - \mathbf{b} \rangle \\
&\leq |\langle \nabla f(\mathbf{a}), \mathbf{a} - \mathbf{b} \rangle| \\
&\leq \sum_{i=1}^{p} |\nabla f(\mathbf{a})_i \, (\mathbf{a}_i - \mathbf{b}_i)| \\
&\leq \|\nabla f(\mathbf{a})\|_p \|\mathbf{a} - \mathbf{b}\|_q,
\end{aligned}
$$

where we used Hölder's inequality for the last inequality. The same argument gives:

$$f(\mathbf{b}) - f(\mathbf{a}) \leq \|\nabla f(\mathbf{b})\|_p \|\mathbf{b} - \mathbf{a}\|_q.$$

Using the absolute value, we can combining the two previous results and obtain the desired inequality. $\square$

The proof of Theorem 3.1 follows from applying Lemma H.1 to the convex function $\mathcal{L}_T$.

## H.2    PROOF OF THEOREM 3.3

We start by introducing some notations. We denote $\mathcal{L}_{\mathbf{x}, y}$ the loss evaluated on a specific data-point $(\mathbf{x}, y) \sim P_T(\mathbf{x}, \mathbf{y})$. We can then decompose the expected loss as $\mathcal{L}_T = \mathbb{E}_{P_T(\mathbf{x}, y)} \mathcal{L}_{\mathbf{x}, y}$. It follows by linearity of the expectation that

$$\nabla \mathcal{L}_T = \mathbb{E}_{P_T(\mathbf{x}, y)} \nabla \mathcal{L}_{\mathbf{x}, y}.$$

Then, we prove the following lemma that gives the formulation of the gradient of the cross-entropy.

**Lemma H.2.** *The gradient of the cross-entropy loss with respect to $\boldsymbol{\omega} = (\mathbf{w}_k)_{k=1}^{K}$ writes*

$$\nabla \mathcal{L}_{\mathbf{x}, y}(\boldsymbol{\omega}) = \left( -y^{(k)} \mathbf{x} (1 - \mathrm{s}_{\boldsymbol{\omega}}^{(k)}(\mathbf{x})) \right)_{k=1}^{K}.$$

*Proof.* First, let's compute the partial derivative of the softmax w.r.t. $\mathbf{w}_k$ for any $k \in \{1, \dots, K\}$. We have:

$$
\begin{aligned}
\frac{\partial \mathrm{s}_{\boldsymbol{\omega}}^{(k)}(\mathbf{x})}{\partial \mathbf{w}_k} &= \frac{\mathbf{x} e^{\mathbf{w}_k^\top \mathbf{x}} \left( \sum_{\tilde{k}} e^{\mathbf{w}_{\tilde{k}}^\top \mathbf{x}} - e^{\mathbf{w}_k^\top \mathbf{x}} \right)}{\left( \sum_{\tilde{k}} e^{\mathbf{w}_{\tilde{k}}^\top \mathbf{x}} \right)^2} \\
&= \mathbf{x} \left( \mathrm{s}_{\boldsymbol{\omega}}^{(k)}(\mathbf{x}) - \left[ \mathrm{s}_{\boldsymbol{\omega}}^{(k)}(\mathbf{x}) \right]^2 \right).
\end{aligned}
$$

Using the chain rule, the partial derivative of the loss w.r.t. $\mathbf{w}_k$ writes:

$$
\begin{aligned}
\frac{\partial \mathcal{L}_{\mathbf{x}, y}(\boldsymbol{\omega})}{\partial \mathbf{w}_k} &= \frac{\partial \mathcal{L}_{\mathbf{x}, y}(\boldsymbol{\omega})}{\partial s(\boldsymbol{\omega}, \mathbf{x})} \cdot \frac{\partial s(\boldsymbol{\omega}, \mathbf{x})}{\partial \mathbf{w}_k} \\
&= \begin{cases} -\frac{1}{\mathrm{s}_{\boldsymbol{\omega}}^{(k)}(\mathbf{x})} \cdot \mathbf{x} \left( \mathrm{s}_{\boldsymbol{\omega}}^{(k)}(\mathbf{x}) - \left[ \mathrm{s}_{\boldsymbol{\omega}}^{(k)}(\mathbf{x}) \right]^2 \right), & \text{if } y^{(k)} = 1 \\ 0, & \text{otherwise} \end{cases} \\
&= -y^{(k)} \mathbf{x} \left( 1 - \mathrm{s}_{\boldsymbol{\omega}}^{(k)}(\mathbf{x}) \right)
\end{aligned}
$$

As the $\frac{\partial \mathcal{L}_{\mathbf{x}, y}(\boldsymbol{\omega})}{\partial \mathbf{w}_k}$ are the coordinates of $\nabla \mathcal{L}_{\mathbf{x}, y}(\boldsymbol{\omega})$, we obtain the desired formulation. $\square$

We now proceed to the proof of Theorem 3.3.

*Proof.* Using the convexity of $\|\cdot\|_p$ and the Jensen inequality, we have that

$$
\begin{aligned}
\|\nabla \mathcal{L}_T(\boldsymbol{\omega})\|_p &= \|\mathbb{E}_{P_T(\mathbf{x},y)} \nabla \mathcal{L}_{\mathbf{x},y}(\boldsymbol{\omega})\|_p \\
&\leq \mathbb{E}_{P_T(\mathbf{x},y)} \|\mathcal{L}_{\mathbf{x},y}(\boldsymbol{\omega})\|_p && \text{(Jensen inequality)} \\
&= \mathbb{E}_{P_T(\mathbf{x},y)} \left( \sum_{i=1}^{D} \sum_{k=1}^{K} |-y^{(k)} \mathbf{x}_i (1 - \mathrm{s}_{\boldsymbol{\omega}}^{(k)}(\mathbf{x})|^p \right)^{1/p} \\
&= \mathbb{E}_{P_T(\mathbf{x},y)} \left( \sum_{k=1}^{K} y^{(k)} \left( 1 - \mathrm{s}_{\boldsymbol{\omega}}^{(k)}(\mathbf{x}) \right)^p \right)^{1/p} \left( \sum_{i=1}^{D} |\mathbf{x}_i^p| \right)^{1/p} \\
&= \mathbb{E}_{P_T(\mathbf{x},y)} \left( \left( 1 - \mathrm{s}_{\boldsymbol{\omega}}^{(k_y)}(\mathbf{x}) \right)^p \right)^{1/p} \left( \sum_{i=1}^{D} |\mathbf{x}_i^p| \right)^{1/p} && (k_y \text{ such that } y^{(k_y)} = 1) \\
&= \mathbb{E}_{P_T(\mathbf{x},y)} \alpha(\boldsymbol{\omega}, \mathbf{x}, y) \|\mathbf{x}\|_p,
\end{aligned}
$$

where $\alpha(\boldsymbol{\omega}, \mathbf{x}, y) = \left( 1 - \mathrm{s}_{\boldsymbol{\omega}}^{(k_y)}(\mathbf{x}) \right)$, with $k_y$ such that $y^{(k_y)} = 1$. We used the fact that $\mathbf{y}$ is a one-hot vector so it has only one nonzero entry. $\qquad \square$

### H.3 PROOF OF REMARK 5.1

*Proof.* Using the reverse Minkowski inequality, as $0 < p < 1$, we have that

$$
\begin{aligned}
\|\boldsymbol{\omega}_s\|_p &= \|\mathbf{c} + \eta \cdot \nabla \mathcal{L}_T(\boldsymbol{\omega}_s)\|_p \geq \|\mathbf{c}\|_p + \eta \cdot \|\nabla \mathcal{L}_T(\boldsymbol{\omega}_s)\|_p \\
&\implies \|\boldsymbol{\omega}_s\|_p - \|\mathbf{c}\|_p \geq \eta \cdot \|\nabla \mathcal{L}_T(\boldsymbol{\omega}_s)\|_p.
\end{aligned}
$$

In the same fashion, we have that

$$
\begin{aligned}
\|\mathbf{c}\|_p &= \|\boldsymbol{\omega}_s - \eta \cdot \nabla \mathcal{L}_T(\boldsymbol{\omega}_s)\|_p \geq \|\boldsymbol{\omega}_s\|_p + \eta \cdot \|\nabla \mathcal{L}_T(\boldsymbol{\omega}_s)\|_p \\
&\implies \|\mathbf{c}\|_p - \|\boldsymbol{\omega}_s\|_p \geq \eta \cdot \|\nabla \mathcal{L}_T(\boldsymbol{\omega}_s)\|_p.
\end{aligned}
$$

We obtain the desired upper bound by combining those results. $\qquad \square$

