# OpenReview forum: "Gradient norm as a powerful proxy to out-of-distribution error estimation"
_ICLR.cc/2024/Conference — Submitted to ICLR 2024_

### Official Review · Reviewer_16Ha · 2023-10-17

**Soundness:** 3 good
**Presentation:** 3 good
**Contribution:** 3 good
**Rating:** 6
**Confidence:** 4

**Summary:**

This paper considered estimating the error in out-of-distribution samples, which is crucial in OOD problems. Specifically, this paper considered a very simple yet effective estimator -- gradient norm w.r.t the down-stream models. I.e, a sample with higher gradient norm indicates an OOD sample. Based on these arguments, theoretical analysis is done in both toy and simple settings. Empirical results clearly support the proposed metric.

**Strengths:**

- In general, I like the proposed idea with simple but well elegant explanations.
- The idea seems novel for me and reasonable in some settings.
- Extensive empirical results

Based on these points, I would recommend a borderline positive.

**Weaknesses:**

- About inherent assumptions. I noticed this paper requires that the model should be calibrated. In general, I would think this assumption may not necessarily be true in several settings. For example, it has been widely proved that deep learning models are poorly calibrated. Based on this, I would suggest (1) additional experiments on the calibration error on current IN-distribution test data (2) additional limitation section about this point.
- About gradient norm detection. Using gradient norm as a detector can be novel in OOD detection. While this might not be sufficiently novel in border distribution shift related papers. For example, paper [1-3] discussed the role of gradient norm/flow in meta-learning, algorithmic fairness and domain adaptation. It can be great if additional discussion is done in border distribution shift related topics.
- About the pseudo labels in the test data. I agree this is generally a non-trivial task because we never know the calibration property of deep neural-network. I was wondering two alternative scenarios (1) if we use random labels (2) we compute the average on all labels.
- Based on the assumption it seems a scoring detector is a sufficient estimator. I.e, a higher gradient score should be OOD and not vice versa (because it depends on the data variance, in your toy example). What do you think about sufficient and necessary conditions in estimating OOD errors?


References

[1] Generalization Bounds For Meta-Learning: An Information-Theoretic Analysis. NeurIPS 21

[2] Fair Representation Learning through Implicit Path Alignment. ICML 22

[3] Algorithm-Dependent Bounds for Representation Learning of Multi-Source Domain Adaptation. AISTAS 23

**Questions:**

See the weakness part. The specific questions and suggestions are provided.

---

> ### Author Response · Authors · 2023-11-17
> **Part 1 of Response to Reviewer 16Ha**
>
> We thank the reviewer for the positive comments and valuable suggestions. We will address the reviewer's concerns below.
>
> **1. Discussion and additional experiments about the model calibration.**
> > About inherent assumptions. I noticed this paper requires that the model should be calibrated. In general, I would think this assumption may not necessarily be true in several settings. For example, it has been widely proved that deep learning models are poorly calibrated. Based on this, I would suggest (1) additional experiments on the calibration error on current IN-distribution test data (2) additional limitation section about this point.
> >
> We thank the reviewer for correctly noticing this possible limitation. We have mentioned earlier that, in theory, the proposed pseudo-labeling strategy depends on how well the prediction probabilities are calibrated. In degraded cases, this can have a negative impact on our approach, e.g., one can imagine a model that outputs only one-hot probabilities with not a high accuracy. However, this it is generally not the case. Indeed, in practice, we do not need to have a perfectly calibrated model as we employ a mixed strategy that assigns pseudo-labels to high-confidence examples and random labels to low-confidence ones. The recent success of applying self-training models to different problems [1, 2, 3] provides evidence of the suitability of the label generation strategy we adopted.
>
> Moreover, when we speak of deep neural networks, which are widely accepted to be poorly calibrated, [4] showed that modern SOTA image models tend to be well-calibrated across distribution shifts.
> To demonstrate it empirically, in the table below, we provide the expected calibration error (ECE, [5]) of ResNet18, one of the considered base models, depending on a difficulty of test data.
> For this, we test first on CIFAR-10 (ID), and then on CIFAR-10C corrupted by brightness across diverse severity from 1 to 5. We can see that ECE is very low for ID data and remains relatively low across all levels of corruption severity, which shows that ResNet is quite well-calibrated on CIFAR-10.
>
> |Severity|ID|1|2|3|4|5|
> | --- | - | - | - | - | - | - |
> |ECE|0.0067|0.0223|0.0230|0.0243|0.0255|0.0339|
>
> On the other hand, in the case of more complex distribution shift like Office-31 data set, we can see in the table below that the calibration error has been increased noticeably. It is interesting to analyze this result together with Figure 3 of the main paper, where we compared the results between the usual pseudo-labeling strategy and the proposed one. Although our method has room for improvement compared to the oracle method, it is also significantly better than "pseudo-labels", indicating that the proposed label generation strategy is less sensitive to the calibration error.
>
> |Domain |DSLR (ID)|Amazon |Webcam |
> | --- | - | - | - |
> |ECE|0.2183|0.2167 |0.4408 |
>
> We thank the reviewer for their insightful suggestions and we added this limitation section in the revised version of the paper.
>
> [1] Sohn, Kihyuk, et al. "Fixmatch: Simplifying semi-supervised learning with consistency and confidence." Advances in neural information processing systems 33 (2020): 596-608.
>
> [2] Dong, Jiahua, et al. "Confident anchor-induced multi-source free domain adaptation." Advances in Neural Information Processing Systems 34 (2021): 2848-2860.
>
> [3] Yu, Yaodong, et al. "Predicting out-of-distribution error with the projection norm." International Conference on Machine Learning. PMLR, 2022.
>
> [4] Minderer, Matthias, et al. "Revisiting the Calibration of Modern Neural Networks." Advances in Neural Information Processing Systems 34 (2021).
>
> [5] Guo, Chuan, et al. "On Calibration of Modern Neural Networks." ICML 34 (2017).
>
> **2. About gradient norm detection.**
> > About gradient norm detection. Using gradient norm as a detector can be novel in OOD detection. While this might not be sufficiently novel in border distribution shift related papers. For example, paper [1-3] discussed the role of gradient norm/flow in meta-learning, algorithmic fairness and domain adaptation. It can be great if additional discussion is done in border distribution shift related topics.
> >
> In Related Work (Appendix B), we have discussed the application of gradient norm in many fields, including OOD detection and generalization error bounds. Basically, GradNorm [1] as an OOD detection method utilizes gradient norm to form a function estimator, where it assumes that the softmax probability of OOD samples should be flatter than that of in-distribution samples. We also discuss the differences between our method and GradNorm [1] in Appendix E. We hope this can address your problem about how gradient norm can be applied in OOD detection.
>
> [1] Huang, Rui, Andrew Geng, and Yixuan Li. "On the importance of gradients for detecting distributional shifts in the wild." Advances in Neural Information Processing Systems 34 (2021): 677-689.

---

> ### Author Response · Authors · 2023-11-17
> **Part 2 of Response to Reviewer 16Ha**
>
> *3. About the pseudo labels in the test data.**
> > About the pseudo labels in the test data. I agree this is generally a non-trivial task because we never know the calibration property of deep neural-network. I was wondering two alternative scenarios (1) if we use random labels (2) we compute the average on all labels.
> >
> (1) *If we only use random labels (RandLabel)?*
>
> We conduct the ablation study with ResNet-18 on 5 datasets. The numerical results of $R^2$ are shown below. From this table, we can observe that if we only use random labels, the performance will degrade drastically. For example, the performance of CIFAR-100 decreases from 0.983 to 0.741.
>
> |Dataset|CIFAR-10|CIFAR-100|TinyImageNet|Entity30|Living17|
> | --- | - | - | - | - | - |
> |RandLabel|0.948|0.741|0.900|0.848|0.881|
> |Ours|**0.971**|**0.987**|**0.971**|**0.970**|**0.949**|
>
> (2) *If we compute the average over all labels (AverLabel)?*
>
> We also conduct the ablation study with ResNet-18 on 5 datasets. The numerical results of $R^2$ are shown below. From this table, we find that our label generation strategy is superior to using average labels. For example, our strategy enhances the performance of AverLabel from 0.897 to 0.971.
>
> |Dataset|CIFAR-10|CIFAR-100|TinyImageNet|Entity30|Living17|
> | --- | - | - | - | - | - |
> |AverLabel|0.949|0.976|0.897|0.837|0.911|
> |Ours|**0.971**|**0.987**|**0.971**|**0.970**|**0.949**|
>
> **4. The sufficient and necessary conditions in estimating OOD errors.**
> > Based on the assumption it seems a scoring detector is a sufficient estimator. I.e, a higher gradient score should be OOD and not vice versa (because it depends on the data variance, in your toy example). What do you think about sufficient and necessary conditions in estimating OOD errors?
> >
> We thank the reviewer for this very interesting question. If one focuses on the motivational example provided in our work, we demonstrated that norm of the gradient is directly linked to the magnitude of the data variance $\sigma_t^x$. An extreme case could be to assume $\sigma_t^x$ very small, say $\sigma_t^x = \varepsilon.$ In this situation, estimating the OOD error is particularly difficult as the norm of the gradient will be scaled down, leading to imperceptible variations between IID and OOD data. Hence, a natural condition to estimate OOD errors could be to consider data with enough variance to avoid degenerated cases such as the one previously mentioned. However, in the general setting, to the best of our knowledge, determining sufficient and necessary conditions on (OOD) error estimation is a hard problem which remains open. We advocate that the analysis needed to answer the question is far from trivial and a real research problem on its own. An example showcasing the complexity of the involved computation is [1], where the authors consider a seemingly simple network for which the analysis is surprisingly very difficult and even requires the usage of the Matlab Symbolic Toolbox to help them simplify the calculation.
>
> We thank the reviewer for their insightful comments. We hope we have adequately addressed their concerns and we will be happy to answer any additional questions. We would be grateful if the reviewer could reconsider the evaluation of our work accordingly.
>
> [1] Arnold, Sébastien M. R., et al. "When MAML Can Adapt Fast and How to Assist When It Cannot", AISTATS 2021, PMLR 130:244-252.

---

### Official Review · Reviewer_8wYw · 2023-10-30

**Soundness:** 2 fair
**Presentation:** 2 fair
**Contribution:** 1 poor
**Rating:** 5
**Confidence:** 3

**Summary:**

This paper introduces an approach that leverages gradients to predict OOD errors. The authors propose a "GRDNORM Score," which quantifies the magnitude of gradients after one gradient step on OOD data. The key idea is that a model should have higher magnitude gradients when it doesn't generalize well to OOD data. The paper provides theoretical insights, demonstrating the effectiveness of this approach through extensive experiments, outperforming state-of-the-art algorithms.

**Strengths:**

The authors support their proposed concept with a thorough set of experiments to validate its efficacy.

**Weaknesses:**

- The problem is not well defined and authors provide more details in supp. material. This way, the reader can not follow the paper well.
- Lack of novelty: In my view, this paper bears a strong resemblance to the work presented in reference [1]. The only difference with [1] in my opinion is using pseudo labels in your method which is not a significant change.

 Furthermore, it is essential to note the existence of another research study in a similar direction, as evidenced by reference [2]. I kindly request that the authors perform a comparative analysis between their paper and these two references, elucidating the primary distinctions and novel contributions of their methodology.



[1] Huang, Rui, Andrew Geng, and Yixuan Li. "On the importance of gradients for detecting distributional shifts in the wild." Advances in Neural Information Processing Systems 34 (2021): 677-689.
[2] Igoe, Conor, et al. "How Useful are Gradients for OOD Detection Really?." arXiv preprint arXiv:2205.10439 (2022).

**Questions:**

I appreciate it if the authors could conduct a comparative analysis of their paper with the references [1] and [2], highlighting the key distinctions and novel aspects of their approach.


[1] Huang, Rui, Andrew Geng, and Yixuan Li. "On the importance of gradients for detecting distributional shifts in the wild." Advances in Neural Information Processing Systems 34 (2021): 677-689.

[2] Igoe, Conor, et al. "How Useful are Gradients for OOD Detection Really?." arXiv preprint arXiv:2205.10439 (2022).

---

> ### Author Response · Authors · 2023-11-17
> **Part 1 of Response to Reviewer 8wYw**
>
> We thank the reviewer for the positive comments and valuable suggestions. We will address the reviewer's concerns below.
>
> **1. Improvement of problem definition**
> > The problem is not well defined and authors provide more details in supp. material. This way, the reader can not follow the paper well.
> >
> We thank the reviewer for pointing out this confusion to us. We added more details to the problem setup explaining what is the exact goal of OOD error estimation problem and its different with OOD detection as explained below in the revised version of our paper.
>
> **2. Relation to GradNorm [1].**
> > Lack of novelty: In my view, this paper bears a strong resemblance to the work presented in reference [1]. The only difference with [1] in my opinion is using pseudo labels in your method which is not a significant change.
> >
> We thank the reviewer for pointing out the potential confusion between our method and GradNorm [1]. In Appendix E, we have discussed in a very detailed manner the differences between the two methods from the view of problem setting, methodology, theoretical insights, and performance. To the best of our knowledge, we are the first to certify the strong linear relationship between gradient norm and test error even under the distribution shift. We reiterate it below.
>
> In general, **OOD error estimation and OOD detection are different problems** with different definitions about OOD. In OOD detection as considered in [1], "OOD" samples refer to those test samples with different label space from the training set, which is known as semantic shift. In OOD error estimation, we use “OOD” to describe examples with covariant shifts or concept shifts, where their ground-truth labels are included in the label set of the training dataset. The goal of OOD detection is to design a binary function estimator to classify whether a coming sample is from in-distribution or not, while the goal of OOD error estimation is to design a score that is correlative to the ground-truth test error. Essentially, OOD detection is a classification problem, while OOD error estimation can be regarded as a linear regression problem.
>
> Due to different problem settings, although both utilize gradient norm, **we capture different information from the gradient space**. GradNorm assumes that the softmax probability of OOD samples is flatter than in-distribution samples, so the magnitude of gradients measures whether the test sample has a flat softmax probability or not. In this case, the magnitude of gradients for OOD samples should be **smaller** than that for in-distribution samples. However, our method actually measures the gradient distance from the source distribution to the target distribution, which means the magnitude of gradients for OOD samples should be **larger** than in-distribution samples. In addition, GradNorm is an **instance-level** function estimator, while our method is a **dataset-level** score.
>
> **Above differences are also reflected in theoretical insights**. Basically, the superiority of GradNorm to other OOD detection methods is that it includes information from both the feature and the output. In our work, we demonstrate that the magnitude of gradients formulates an upper bound on the true OOD error (Corollary 3.2), showcasing the theoretical soundness of our method for OOD error estimation.
>
> **Empirical Comparison.**  We also numerically compared our method with [1], and the results obtained ($R^2$) with ResNet18 are shown in Table 4 of the revised version of the paper (table below). We can observe from this table that our method is superior to GradNorm for OOD error estimation across diverse distribution shifts. For example, our method outperforms GradNorm on TinyImageNet with a large margin from 0.894 to 0.971. It shows that **GradNorm is not suitable for OOD error estimation problems**.
>
> |Method|CIFAR-10|CIFAR-100|TinyImageNet|Office-31|Office-Home|Entity-30|Living-17|
> | --- | - | - | - | - | - | - | - |
> |GradNorm [1]|0.951|0.978|0.894|0.596|0.848|0.964|0.942|
> |Ours|**0.971**|**0.987**|**0.971**|**0.675**|**0.876**|**0.970**|**0.949**|
>
> [1] Huang, Rui, Andrew Geng, and Yixuan Li. "On the importance of gradients for detecting distributional shifts in the wild." Advances in Neural Information Processing Systems 34 (2021): 677-689.

---

> ### Author Response · Authors · 2023-11-17
> **Part 2 of Response to Reviewer 8wYw**
>
> **3.Relation to ExGrad [2].**
> >Furthermore, it is essential to note the existence of another research study in a similar direction, as evidenced by reference [2]. I kindly request that the authors perform a comparative analysis between their paper and these two references, elucidating the primary distinctions and novel contributions of their methodology.
> >
> We thank the reviewer for pointing out to the interesting ExGrad method [2]. This paper tackles the OOD detection problem as in [1] and proposes an instance-level score based on a similar decomposition in terms $U$ and $V$ as in [1]. As such, all our explanations provided above about GradNorm also apply to ExGrad.
>
> To compare their performance, we also conduct an experiment on 3 datasets with ResNet-18, which results of $R^2$ are shown below.
>
> |Dataset|CIFAR-10|Entity30|Living17|
> | --- | - | - | - |
> |ExGrad|0.198|0.205|0.018|
> |Ours|**0.971**|**0.987**|**0.971**|**0.970**|**0.949**|
>
> From the above table, we find that ExNorm is completely invalid under the OOD error estimation task. This phenomenon is because ExNorm can be only deployed with the batch size as 1, from which we obtain imprecise gradient-norm information about the whole dataset.
>
> We hope we have adequately addressed the reviewers' concerns and we will be happy to answer any additional questions. If the reviewer finds our answer satisfactory, we would be grateful if they can reconsider their score accordingly.
>
> [1] Huang, Rui, Andrew Geng, and Yixuan Li. "On the importance of gradients for detecting distributional shifts in the wild." Advances in Neural Information Processing Systems 34 (2021): 677-689.
>
> [2] Igoe, Conor, et al. "How Useful are Gradients for OOD Detection Really?." arXiv preprint arXiv:2205.10439 (2022).

---

> ### Comment · Reviewer_8wYw · 2023-11-21
>
> Thanks for providing the rebuttal!
> I have no question any more and would consider your rebuttal in my final decision/score.

---

> > ### Author Response · Authors · 2023-11-21
> > **Thank you**
> >
> > We would like to thank the reviewer for their feedback. We would greatly appreciate it if the reviewer could let us know whether the provided arguments may allow them to reconsider their current score.

---

### Official Review · Reviewer_XBio · 2023-10-31

**Soundness:** 3 good
**Presentation:** 3 good
**Contribution:** 3 good
**Rating:** 6
**Confidence:** 4

**Summary:**

In this work, the authors focus on the problem of Out-of-distribution (OOD) Error Estimation, which aims to estimate the test performances under distribution shifts in an unsupervised manner. Inspired by the relationship between the gradient information and the generalization ability within the deep neural network, the authors proposed a novel approach to solve the OOD error estimation problem by leveraging the magnitude of the gradient. Specifically, by analyzing the fine-tuning process of a pre-trained model on the new test data, the authors showed that the gradient information is crucial for OOD error prediction. Then, a novel statistic, named GRDNORM Score, was proposed by calculating the vector norm of the gradient of the last layer through one-step backpropagation, under a confidence-based pseudo-labeling policy. With the empirical evaluations on different datasets, network architectures, and distribution shifts, the proposed method showed its effectiveness and efficiency for OOD error prediction.

**Strengths:**

1. The problem that this work investigated is interesting and valuable. OOD error estimation provides a novel perspective to study how the model performs under the distribution shifts, but without accessing the ground-truth test labels.
2. The usage of gradient information to perform OOD error prediction is novel. To the best of my knowledge, the role of gradient information was less explored in the field. This work provides another viewpoint to build the relationship between the OOD performance and test data, under a data-centric perspective. The proposed method also showed the advantage of the computational overhead.
3. The motivation is clear and reasonable. The theoretical analysis and empirical verifications are promising and smooth.
4. The experiments are extensive. The proposed method is verified under different datasets, distributional shifts, etc.
5. The presentation is clear. This manuscript is well-written.

**Weaknesses:**

1. Some steps of the proposed highly relied on some hyperparameters, such as the confidence threshold adopted for the pseudo-labeling in Eq.(5). However, the manuscript did not provide a detailed explanation about the choice of this hyperparameter.
2. It seems that an important baseline is missing in the comparisons with the proposed method.

See the Questions part.

**Questions:**

1. Discussion about the pseudo-label generation process. In Eq. (5), the authors proposed to apply a threshold $\tau$ to control the confidence-based label generation process. In my understanding, if we set too small values for this hyperparameter, it seems we will assign a determined pseudo-label even for less confident samples. In contrast, using too large values for $\tau$ can be viewed as discarding more information in this process. Here are my questions:

    - 1.1. How did the authors choose a proper threshold $\tau$ in this process? I did not find any discussion or analysis for this in the main body or the appendix. But I believe the choice of this threshold does matter for the final performance.

    - 1.2 About the low-confidence case. In Eq.(5), if the maximum probability predicted by the model is still lower than a threshold $\tau$, the authors proposed to assign a random label from the label space. Is this process reasonable and can it be replaced by other methods? For example, if we set $\tau=0.5$ and a sample is predicted with  $f_{\theta} (\mathbf{x}_{i})=[0.1,0.4,0.35,0.15]$  in a four-way classification scenario. If we adopt the label-generation strategy in this paper, we should randomly generate pseudo-label from the label space $\mathcal{Y}=${0,1,2,3}.

However, even though the maximum probability $0.4< \tau=0.5$, we can still observe that the model tends to predict $\mathbf{x}_{i}$ into Class {1,2} with higher probabilities. Thus, will we discard too much information if we naively select a random label within the whole label space? And I guess there are other ways to deal with low-confidence samples. For example:

(a) randomly generate pseudo labels from the top-$K$ largest confidences, e.g., generating from $\mathcal{Y}^{\prime}=${1,2} rather than the full label space $\mathcal{Y}=${0,1,2,3};

(b) directly adopt the pseudo label $[0.1,0.4,0.35,0.15]$ for this low-confident sample.

2. It seems that a recent work [1] was missed in the comparisons between the proposed method and the baselines. In that work, the confidence and the disperity of the prediction matrix on the test dataset were considered for OOD error estimation and the nuclear norm was adopted to predict the OOD error. Could the authors provide comparisons between the proposed method and this work, both in terms of estimation performance and computational efficiency?

3. Some typos. For example:
- After Equation (2),  $\mathbf{s}_{k}$ should be  $\mathbf{s}_{w}^{k}$ be for a consistent expression;
- Before Equation (3), $\mathcal{D}_{test}=\{\tilde{\mathbf{x}}\}_{i=1}^{m}$ should be $\mathcal{D}_{test}=\{\tilde{\mathbf{x}_{i}}\}_{i=1}^{m}$

References:

[1] Deng et al. Confidence and Dispersity Speak: Characterizing Prediction Matrix for
Unsupervised Accuracy Estimation. International Conference on Machine Learning, 2023.

---

> ### Author Response · Authors · 2023-11-17
> **Part 1 of Response to Reviewer XBio**
>
> We thank the reviewer for the positive comments and valuable suggestions. We will address the reviewer's concerns below.
>
> **1. Choice of a proper threshold $\tau$ on the final performance?**
> > 1.1. How did the authors choose a proper threshold
>  in this process? I did not find any discussion or analysis for this in the main body or the appendix. But I believe the choice of this threshold does matter for the final performance.
> >
>
> In this work, we set the value of $\tau$ as 0.5 across all datasets and network architectures. To demonstrate the impact of threshold $\tau$ on the final performance, we conduct an ablation study on CIFAR-10 with ResNet18 by changing the value of $\tau$. The results are shown below. From this table, we can observe that the final performance firstly experiences a slight increase, and achieves the top when $\tau$ is 0.4 and 0.5. Then it decreases with the increase of $\tau$.
>
> |Threshold|0.0|0.1|0.2|0.3|0.4|0.5|0.6|0.7|0.8|0.9|
> | --- | - | - | - | - | - | - | - | - | - | - |
> |Ours|0.963|0.963|0.964|0.965|**0.971**|**0.971**|0.967|0.962|0.963|0.959|
>
> In this work, we also set $\tau$ as 0.5 due to an intuitive reason. If a label contains a softmax probability below 0.5, it means that this predicted label has over 50% probability to be wrong. It means that this label has higher probability to be incorrect than to be correct. Thus, we tend to regard it as an incorrect prediction.
>
>
> **2. Other strategies for label assignment of low-confidence samples.**
>
> > (a) Randomly generate pseudo labels from the top-K largest confidences, e.g., generating from $\mathcal{Y}' = \{1,2\}$ rather than the full label space $\mathcal{Y} = \{0,1,2,3\}$;
> >
> In this experiment, we replace the whole label set used in our label generation strategy with that from the top-k largest confidences. The set of top-k largest confidences is determinded by its accumulative probability which is required to be at least 90%. The results of $R^2$ with ResNet-18 are shown below.
>
> |Dataset|CIFAR-10|CIFAR-100|TinyImageNet|Entity30|Living17|
> | --- | - | - | - | - | - |
> |Top-K|0.968|0.986|0.967|**0.977**|0.940|
> |Ours|**0.971**|**0.987**|**0.971**|0.970|**0.949**|
>
> From the above table, we can see that the two label generation strategies are comparable based on their performance.
>
> > (b) Directly adopt the pseudo label [0.1, 0.4, 0.35, 0.15] for this low-confident sample.
> >
> We have discussed this scenario in the second ablation study, where "Entropy" in Figure 4 shows the corresponding results. It shows that though adopting softmax probability for low-confidence samples enhances the performance under the synthetic shift and the novel subpopulation shift, it struggles under the natural shift.
>
> The next tables illustrate the numerical results of $R^2$ with diverse network architectures and datasets. We can see that, while entropy is comparable, sometimes better, to our approach when the shift is artificial, this method heavily struggles under natural shift (Office-31, Office-Home). On average, our method enables higher performance and is more robust.
>
> |ReNet-18|CIFAR-10|CIFAR-100|TinyImageNet|Office-31|Office-Home|Entity30|Living17|Average|
> | --- | - | - | - | - | - |  - |  - |  - |
> |Entropy|**0.990**|**0.990**|**0.984**|0.404|0.802|**0.985**|**0.958**|0.874|
> |Ours|0.971|0.987|0.971|**0.675**|**0.876**|0.970|0.949|**0.914**|
>
> |ReNet-50|CIFAR-10|CIFAR-100|TinyImageNet|Office-31|Office-Home|Entity30|Living17|Average|
> | --- | - | - | - | - | - |  - |  - |  - |
> |Entropy|**0.992**|**0.996**|**0.992**|0.400|0.799|**0.987**|**0.953**|0.875|
> |Ours|0.969|0.991|0.980|**0.604**|**0.829**|0.957|0.931|**0.895**|
>
> |WRN-50-2|CIFAR-10|CIFAR-100|TinyImageNet|Office-31|Office-Home|Entity30|Living17|Average|
> | --- | - | - | - | - | - |  - |  - |  - |
> |Entropy|**0.992**|0.996|**0.990**|**0.758**|0.357|**0.977**|**0.924**|0.856|
> |Ours|0.971|**0.997**|0.976|0.694|**0.809**|0.949|0.910|**0.901**|

---

> > ### Comment · Reviewer_XBio · 2023-11-21
> > **Further comments after the authors' response**
> >
> > Thank you for your reply. I appreciate the effort from the authors in answering our questions with further explanations and additional empirical verifications. Some of my concerns have been addressed. And I will carefully consider the provided rebuttal during the AC-Reviewer discussion period.

---

> > > ### Author Response · Authors · 2023-11-21
> > > **Thank you for your feedback**
> > >
> > > We thank the reviewer for reaching out to us and notifying us about their appreciation. If some of the concerns are still unaddressed, please do let us know as soon as possible and we will do our best to clear out any further confusion.
> > >
> > > If our current answer is satisfactory, we would greatly appreciate it if the reviewer could consider changing their score to reflect it.

---

> > > > ### Comment · Reviewer_XBio · 2023-11-22
> > > > **Comments from Reviewer XBio (2)**
> > > >
> > > > Thanks for your reply. Based on the responses provided by the author and the discussion with other reviewers, I have some further comments:
> > > > - Additional results about choosing threshold $\tau$.
> > > >
> > > > I noticed that when you set $\tau$ for extreme cases ($0.0$ or $0.9$), the final performances on CIFAR10 are still comparable with the case $\tau=0.5$ (with a drop of only $0.8\%-1.2\%$). In your paper, you assumed that the classifier made mistakes mostly on data with low prediction confidence (i.e., smaller than $\tau$), for which you deliberately assigned noisy pseudo-labels. In my perspective, when setting $\tau=0.0$, we fully trust the pre-trained model and directly adopt the maximum prediction as the pseudo-class; if we set $\tau=0.9$, we only trust predictions with very very high maximum probability and assign random labels for any samples whose maximum probability is even slightly lower than $0.9$. It is intuitively weird to see that the final performance did not vary too much within such two extreme cases. I understand that perhaps CIFAR-10 is an easy dataset. However, I wonder whether this phenomenon indicated that we require the pre-trained model a well-calibrated one.
> > > >
> > > > - Comparisons with NuclearNorm
> > > >
> > > > Thanks for providing such a comparison. I understand that comparing with a very recently published paper is a hard job. Based on your results, it seems that the proposed method does not have too many advantages over NuclerNorm, especially concerning computational efficiency. For example, on CIFAR10/CIFAR100/Entity30/Living17 datasets, it seems NuclearNorm has better performances as well as smaller time consumption.

---

> ### Author Response · Authors · 2023-11-17
> **Part 2 of Response to XBio**
>
> **3. Comparison with Nuclear Norm [1].**
> > 2. It seems that a recent work [1] was missed in the comparisons between the proposed method and the baselines. In that work, the confidence and the disperity of the prediction matrix on the test dataset were considered for OOD error estimation and the nuclear norm was adopted to predict the OOD error. Could the authors provide comparisons between the proposed method and this work, both in terms of estimation performance and computational efficiency?
>
>
> We thank the reviewer for point out this this very recent state-of-the-art method. As per reviewer's request, we compare the two methods with ResNet-18 on 6 datasets representing three kinds of distribution shifts, which results of both performance ($R^2$) and computational time ($T$) are shown below. From the tables, we see that our method is much more robust to different types of shift: while being slightly worse under mild shift scenarios (CIFAR datasets, Entity and Living17, ~2% drop), it provides an impressive performance boost under natural shift (Office31 and Office-Home, >10% increase).
>
> We note, however, that given how recent is this publication (less than two months at the moment of our submission), we would like to ask the reviewer to be indulgent regarding our comparison to [1] as we didn't have time to do it more rigorously.
>
>
> Meanwhile, since Nuclear Norm is training-free but our method is based on self-training, it is naturally more time-consumming than Nuclear Norm.
>
> |$R^2$|CIFAR-10|CIFAR-100|Office-31|Office-Home|Entity30|Living17|Average|
> | --- | - | - | - | - | - |  - |  - |
> |Nuclear|**0.995**|**0.996**|0.547|0.785|**0.988**|**0.978**|0.882|
> |Ours|0.971|0.987|**0.675**|**0.876**|0.970|0.949|**0.904**|
>
> |$T$|CIFAR-10|CIFAR-100|Office-31|Office-Home|Entity30|Living17|
> | --- | - | - | - | - | - |  - |
> |Nuclear|**7.523**|**8.311**|13.704|**5.468**|**5.253**|**1.871**|
> |Ours|47.207|47.174|**4.475**|14.559|32.882|32.660|
>
> **4. Typos**
> > 3. Some typos. For example: After Equation (2), $\mathbf{s}^{k}$ should be $\mathbf{s}_{w}^{k}$ be for a consistent expression; Before Equation (3), $\mathcal{D}_{test}=\{\tilde{\mathbf{x}}\}_{i=1}^{m}$ should be $\mathcal{D}_{test}=\{\tilde{\mathbf{x}_{i}}\}_{i=1}^{m}$
>
> We thank the reviewer for pointing out to these typos and we corrected them in the revised version of our paper.
>
> We would be grateful if the reviewer could reconsider the evaluation of our work given all the replies above and in case we do not manage to either reproduce the results of [1] or do a full-scale comparison.
>
> [1] Deng et al. Confidence and Dispersity Speak: Characterizing Prediction Matrix for Unsupervised Accuracy Estimation. International Conference on Machine Learning, 2023.

---

> ### Author Response · Authors · 2023-11-22
> **Thanks for the valuable comments and additional clarification for Reviewer XBio**
>
> We thank the reviewer for his prompt answer and address their concerns below.
>
> **1. Additional results about choosing threshold $\tau$.**
>
> We are sorry for any confusion about this experience. To clarify, we conducted an additional experiment on Office-31 with ResNet-18 to show the model requires good calibration, especially for more complex datasets. The results are shown below. We find that the top performance is achieved near $\tau=0.5$, and the drop for the extreme cases is clearer.
>
> |Threshold|0.0|0.1|0.2|0.3|0.4|0.5|0.6|0.7|0.8|0.9|
> | --- | - | - | - | - | - | - | - | - | - | - |
> |Ours|0.495|0.498|0.532|0.674|**0.685**|0.667|0.545|0.451|0.114|0.131|
>
>
> **2. Comparisons with Nuclear Norm**
>
> We appreciate the reviewer's introduction of Nuclear Norm to us. However, despite its slight improvement on synthetic shifts, Nuclear Norm suffers from a dramatic drop in performance under natural shifts (Office-31 and Office-Home), which severely diminishes its robustness. Contrary to Nuclear Norm, our method balances well across different distribution shifts and remains better on average. It means our method is safer in the real world, especially when the type of the test distribution shift is unknown. The same conclusion stands concerning our second ablation study, where we adopt softmax probability for low-confidence samples. We believe that the robustness of our method, as well as its strong performance on a wide range of baselines and models, gives it several advantages compared to its counterparts, including Nuclear Norm.
>
> As for computational efficiency, compared to training-free methods, our approach requires a reasonable amount of time while ensuring robustness and delivering enhanced performance. Compared to other self-training methods (ProjNorm), our method dramatically decreases their computational cost and enhances their performance.
>
> We thank again the reviewer for your insightful feedback and interesting questions. We hope we have addressed your concerns and will be happy to answer any additional questions. If you find our answers satisfactory, we hope the reviewer will reconsider the evaluation of our work in this sense.

---

### Official Review · Reviewer_b4cf · 2023-10-31

**Soundness:** 2 fair
**Presentation:** 3 good
**Contribution:** 2 fair
**Rating:** 5
**Confidence:** 4

**Summary:**

The paper explores the use of gradient magnitude from the classification layer as an indicator of Out-of-Distribution (OOD) data, acquired via backpropagation from the cross-entropy loss following a single gradient step. The primary notion is that the model should be fine-tuned with greater gradient magnitudes when it struggles to generalize to the OOD dataset. Empirical evidence further validates this concept.

**Strengths:**

S1: The paper considers using the magnitude of gradients from the classification layer as OOD indicator, obtained through backpropagation from the cross-entropy loss following a single gradient step on Out-of-Distribution (OOD) data.

S2: The main concept revolves around the notion that the model needs to be calibrated with larger gradient magnitudes in cases where it fails to generalize to the OOD dataset. Empirical evidences also supports the idea.

**Weaknesses:**

1. I am afraid the paper seems to have significant overlap with the published paper in terms of the idea of using parameterization norm as a measurement: "[A] Rui Huang et al., On the Importance of Gradients for Detecting Distributional Shifts in the Wild (GradNorm)", who both consider using the gradients norm of the parameters as an indicator of OOD data. Surprisingly, the name of the approches are even the same (GradNorm). The only difference is that this paper under review is considering backpropogating vanilla softmax loss, under which the parameterization norm is computed, whereas in paper [A], an KL is computed. But this is very minor difference, as both sotfmax and KL are just distinguishes in how the distribution discrepancy is measured. I am afraid this significantly limits the novelty of the paper.  Please compare the proposed method with [A].

2. Please compare empirically with the mentioned paper and illustrates why the proposed method should have any capacity to be more advantageous than [A].

[A] Rui Huang et al., On the Importance of Gradients for Detecting Distributional Shifts in the Wild (GradNorm)

**Questions:**

Please see above for the questions to be addresed. Please correct me during rebuttal, if there is any misunderstanding here.

---

> ### Author Response · Authors · 2023-11-17
> **Response to Reviewer b4cf**
>
> We thank the reviewer for the positive comments and valuable suggestions. We will address the reviewer's concerns below.
>
> **1. Relation to GradNorm [1].**
> > 1. I am afraid the paper seems to have significant overlap with the published paper in terms of the idea of using parameterization norm as a measurement: "[A] Rui Huang et al., On the Importance of Gradients for Detecting Distributional Shifts in the Wild (GradNorm)", who both consider using the gradients norm of the parameters as an indicator of OOD data. Surprisingly, the name of the approches are even the same (GradNorm). The only difference is that this paper under review is considering backpropogating vanilla softmax loss, under which the parameterization norm is computed, whereas in paper [A], an KL is computed. But this is very minor difference, as both sotfmax and KL are just distinguishes in how the distribution discrepancy is measured. I am afraid this significantly limits the novelty of the paper. Please compare the proposed method with [A].
> >
> We thank the reviewer for pointing out the potential confusion between our method and GradNorm [1]. In Appendix E, we have discussed in a very detailed manner the differences between the two methods from the view of problem setting, methodology, theoretical insights, and performance. To the best of our knowledge, we are the first to certify the strong linear relationship between gradient norm and test error even under the distribution shift. We reiterate it below.
>
> In general, **OOD error estimation and OOD detection are different problems** with different definitions about OOD. In OOD detection as considered in [1], "OOD" samples refer to those test samples with different label space from the training set, which is known as semantic shift. In OOD error estimation, we use “OOD” to describe examples with covariant shifts or concept shifts, where their ground-truth labels are included in the label set of the training dataset. The goal of OOD detection is to design a binary function estimator to classify whether a coming sample is from in-distribution or not, while the goal of OOD error estimation is to design a score that is correlative to the ground-truth test error. Essentially, OOD detection is a classification problem, while OOD error estimation can be regarded as a linear regression problem.
>
> Due to different problem settings, although both utilize gradient norm, **we capture different information from the gradient space**. GradNorm assumes that the softmax probability of OOD samples is flatter than in-distribution samples, so the magnitude of gradients measures whether the test sample has a flat softmax probability or not. In this case, the magnitude of gradients for OOD samples should be **smaller** than that for in-distribution samples. However, our method actually measures the gradient distance from the source distribution to the target distribution, which means the magnitude of gradients for OOD samples should be **larger** than in-distribution samples. In addition, GradNorm is an **instance-level** function estimator, while our method is a **dataset-level** score.
>
> **The above differences are also reflected in theoretical insights**. Basically, the superiority of GradNorm to other OOD detection methods is that it includes information from both the feature and the output. In our work, we demonstrate that the magnitude of gradients formulates an upper bound on the true OOD error (Corollary 3.2), showcasing the theoretical soundness of our method for OOD error estimation.
>
> **2. Empirical Comparison.**
> > 2. Please compare empirically with the mentioned paper and illustrate why the proposed method should have any capacity to be more advantageous than [1].
>
> We also numerically compared our method with [1], and the results obtained ($R^2$) with ResNet18 are shown in Table 4 of the revised version of the paper (table below). We can observe from this table that our method is superior to GradNorm for OOD error estimation across diverse distribution shifts. For example, our method outperforms GradNorm on TinyImageNet with a large margin from 0.894 to 0.971. It shows that **GradNorm is not suitable for OOD error estimation problems**.
>
> |Method|CIFAR-10|CIFAR-100|TinyImageNet|Office-31|Office-Home|Entity-30|Living-17|
> | --- | - | - | - | - | - | - | - |
> |GradNorm [1]|0.951|0.978|0.894|0.596|0.848|0.964|0.942|
> |Ours|**0.971**|**0.987**|**0.971**|**0.675**|**0.876**|**0.970**|**0.949**|
>
> We thank the reviewer for their insightful comments. We hope we have adequately addressed their concerns and we will be happy to answer any additional questions. We would be grateful if the reviewer could reconsider the evaluation of our work accordingly.
>
> [1] Huang, Rui, Andrew Geng, and Yixuan Li. "On the importance of gradients for detecting distributional shifts in the wild." Advances in Neural Information Processing Systems 34 (2021): 677-689.

---

> ### Comment · Reviewer_b4cf · 2023-11-22
> **Thanks for your response.**
>
> Thanks for the response from the authors. However, even if the method actually measures the gradient distance from the source distribution to the target distribution (rather than the used uniform distribution as in GradNorm), the basic fundamental loss discrepancy still is similar to the gradient computation mentioned in GradNorm. This has shown limited novelty of the method in terms of how loss is defined. The reason why OOD detection (as in GradNorm) uses uniform distribution is because of the OOD detection task assumes OOD samples should be agnostic to in-liners. But I am afraid this does not change the fact that the two (this work under the setting of OOD error estimation and GradNorm) are using basically same measurements, i.e., gradient as the core metric to threshold OOD data. I am afraid I will have to maintain my score.

---

> > ### Author Response · Authors · 2023-11-22
> > **Thanks for the feedback and additional clarification for Reviewer b4cf**
> >
> > We thank the reviewer for their answer. We would like to highlight several points to clear out any confusion for the reviewer and emphasize that differences between our method and GradNorm [1] are not limited to the use of the target distribution instead of the uniform one.
> >
> > 1) **Methodology and Theory.** The critical differences between OOD detection and OOD error estimation lead us to develop a methodology that is different both in *model design and theoretical soundness*. While GradNorm [1] uses an *instance-level* gradient score to threshold OOD data, **this is not the case with our method**. Instead, we rely on the gradient at the *dataset level* to give a proxy of the true test error. To do so, we proposed a new label generation strategy that empirically outperforms both naive pseudo-labeling and random labeling. Finally, we demonstrated the theoretical soundness of our method by showing that the magnitude of the gradient formulates an upper bound on the true OOD error.
> >
> > 2) **Experimental Evidence.** In our answer to the reviewer, we explained in detail why gradient-based OOD detection methods are not suitable for OOD error estimation and showed that GradNorm [1] and Exgrad [2] are **subpar experimentally** as well. We believe that the obtained improvements over GradNorm and ExGrad are **significant (>3% on average, >7% for natural shift over GradNorm, >80% on average over ExGrad)**.
> >
> > In light of all these explanations and the additional experiments we conducted, we are surprised that the reviewer reiterates the same arguments and hope to have made the main contributions and novelty of our work clearer.
> >
> > To find common ground with the reviewer, we would like to propose changing the narrative of our work and describe in detail the failure cases of the gradient-based OOD detection methods and put a particular emphasis -- both theoretically and empirically -- on how our proposal allows us to deal with natural shift while also bringing an improvement in other scenarios with a milder distribution shift.
> >
> > We thank again the reviewer for their constructive remarks and remain at their disposal in case such a change would suit them more. We hope that our answer will help the reviewer reconsider the evaluation of our work.
> >
> > [1] Huang, Rui, Andrew Geng, and Yixuan Li. "On the importance of gradients for detecting distributional shifts in the wild." Advances in Neural Information Processing Systems 34 (2021): 677-689.
> >
> > [2] Igoe, Conor, et al. "How Useful are Gradients for OOD Detection Really?." arXiv preprint arXiv:2205.10439 (2022).

---

### Author Response · Authors · 2023-11-17
**General Comment and Summary of Updates to Manuscript**

We thank the reviewers for noting that we address an important problem (Reviewer XBio), with both an interesting theoretical analysis (Reviewer XBio, Reviewer 8wYw, Reviewer 16Ha) and strong empirical evidence (Reviewer XBio, Reviewer b4cf, Reviewer 8wYw, Reviewer 16Ha).

Apart from additional experimental evaluations, the reviewers raised a common concern about the relationship of our approach to OOD detection methods, such as GradNorm [1] and ExGrad [2].

> Comparison to OOD detection methods, such as GradNorm [1] and ExGrad [2]

**Main differences between OOD detection and OOD error estimation**

OOD error estimation considered in our work and OOD detection considered in both [1] and [2] are **different topics** with different definitions regarding the concept of OOD. In OOD detection as considered in [1] and [2], “OOD” samples refer to those test samples with different label space from the training set (model trained on digits and tested on cats), which is known as semantic shift. In OOD error estimation, we use “OOD” to describe examples with covariant shifts or concept shifts, where their ground-truth labels are included in the label set of the training dataset. The goal of OOD detection is to design a binary function estimator to classify whether a coming sample is from in-distribution or not, while the goal of OOD error estimation is to design a score that is correlative to the ground-truth test error. Essentially, OOD detection is a classification problem ([1] and [2] are evaluated with respect to AUROC), while OOD error estimation can be regarded as a linear regression problem (evaluated with respect to $R^2$ and $\rho$). We summarize those differences in the table below.

|Learning Problem|Goal|Scope|Metric|
| --- | - | - | - |
| OOD detection| Predict ID/OOD | $\tilde{\mathbf{x}}_i$ | AUROC |
| OOD error estimation| Proxy to test error | $\mathcal{D}_{\text{test}}$ | $R^2$ and $\rho$ |

**Comparison to GradNorm [1] and ExGrad [2]**

Our paper already provided such a discussion in Appendix E and a favorable comparison to [1]. For the sake of clarity, we now extended this discussion in the revised version of our paper (in Appendix E due to space constraints). We also added a remark (Remark 2.1, extended in Remark B.1) in the problem setup highlighting the difference between the two problem setups.

**Summary of our changes, in blue in the revised version of our draft.**
- Additional discussion on the differences between OOD error estimation and OOD detection in Remark 2.1 (main paper) and Remark B.1 (Appendix B) (**Reviewers b4cf and 8wYw**)
- Additional discussion on the choice of proper threshold, with additional experiments, in Appendix F (**Reviewer XBio**).
- Additional discussion on the model calibration, with additional experiments, in Appendix G (**Reviewer 16Ha**).
- Rewriting of Section 2 with details on OOD error estimation (**Reviewer 8wYw**)
- Additional discussion and experiments for comparison with GradNorm [1] in Appendix E (**Reviewers b4cf and 8wYw**)
- Correction of typos (**Reviewer XBio**)

We hope that we have adequately addressed the reviewers' concerns and we will be happy to answer any additional questions. We hope that this, together with the multiple additional experiments provided in the individual answers, will allow the reviewers to reconsider their evaluation of our work.

[1] Huang, Rui, Andrew Geng, and Yixuan Li. "On the importance of gradients for detecting distributional shifts in the wild." Advances in Neural Information Processing Systems 34 (2021): 677-689.

[2] Igoe, Conor, et al. "How Useful are Gradients for OOD Detection Really?." arXiv preprint arXiv:2205.10439 (2022).

---

### Meta-Review · Area_Chair_BUjJ · 2023-12-12

**Metareview:**

Th paper uses Gradient Norm for estimating Out of Distribution(OOD) Error. The author(s) argue that while there is existing literature on use of gradients in detecting OOD, there is no work in estimating the OOD error. This is a subtle but a critical point which is key to establishing the novelty of the work.
Given that ICLR is a very competitive fora it is important the
paper have more detailed discussion on the difference between previous gradient based approaches  for OOD  detection versus OOD error estimation. The discussion should establish that there are considerable divergences from existing work meriting a separate investigation. A more detailed discussion in the paper would have  helped in making a stronger case for acceptance of the paper.

**Justification For Why Not Higher Score:**

The novelty of the work appears to be limited. Given that ICLR is a very competitive fora it is important the
paper have more detailed discussion on the difference between previous approaches which uses it for OOD  detection versus OOD error estimation. The discussion should establish that there are considerable divergences from existing work meriting a separate investigation.

**Justification For Why Not Lower Score:**

N/A

---

### Decision · Program_Chairs · 2024-01-16

Reject